# Bacteria exposed to antiviral drugs develop antibiotic cross-resistance and unique resistance profiles

Veronica J. Wallace[1], Eric G. Sakowski[1,2], Sarah P. Preheim[1] & Carsten Prasse [1✉]

Antiviral drugs are used globally as treatment and prophylaxis for long-term and acute viral infections. Even though antivirals also have been shown to have off-target effects on bacterial growth, the potential contributions of antivirals to antimicrobial resistance remains unknown. Herein we explored the ability of different classes of antiviral drugs to induce antimicrobial resistance. Our results establish the previously unrecognized capacity of antivirals to broadly alter the phenotypic antimicrobial resistance profiles of both gram-negative and gram-positive bacteria *Escherichia coli* and *Bacillus cereus*. Bacteria exposed to antivirals including zidovudine, dolutegravir and raltegravir developed cross-resistance to commonly used anti-biotics including trimethoprim, tetracycline, clarithromycin, erythromycin, and amoxicillin. Whole genome sequencing of antiviral-resistant *E. coli* isolates revealed numerous unique single base pair mutations, as well as multi-base pair insertions and deletions, in genes with known and suspected roles in antimicrobial resistance including those coding for multidrug efflux pumps, carbohydrate transport, and cellular metabolism. The observed phenotypic changes coupled with genotypic results indicate that bacteria exposed to antiviral drugs with antibacterial properties in vitro can develop multiple resistance mutations that confer cross-resistance to antibiotics. Our findings underscore the potential contribution of wide scale usage of antiviral drugs to the development and spread of antimicrobial resistance in humans and the environment.

[1] Department of Environmental Health and Engineering, Johns Hopkins University, Baltimore, MD, USA. [2] Department of Science, Mount St. Mary's University, Emmitsburg, MD, USA. ✉email: cprasse1@jhu.edu

Antiviral drugs are used worldwide to treat viral diseases that affect millions of human lives. Antivirals attenuate viral infection by inhibiting a virus's ability to replicate, often by targeting the proteins or enzymes used by a virus to infect, multiply, or release new viral particles from a host[1]. Ninety antivirals are currently (as of 2017) approved by the Food and Drug Administration in the United States, including single-compound and combination antivirals for influenza virus, hepatitis B and C, herpes simplex virus 1 and 2, and human immunodeficiency virus (HIV)[2]. Worldwide, an estimated 38 million people are infected with HIV[3]. Three hundred twenty-five million people are living with hepatitis B and/or C[4], and over 3.7 billion cases of herpes simplex virus type 1 exist in individuals under the age of 50 alone[5]. An estimated three to five million severe cases of influenza virus infection occur annually worldwide[6]. As the number of cases of viral diseases increases every year, the use of antiviral drugs is expected to rise as well[7]. Some viral infections such as seasonal influenza may be treated with only short-term antiviral therapy; others such as HIV/AIDS can require long-term to lifetime antiviral treatment. In addition, the on-going SARS-CoV-2 pandemic adds to the list of global viral diseases treated with antiviral drugs[8].

Given the extensive use of antiviral drugs worldwide, the unforeseen consequences of antiviral drugs must also be considered. Antiviral drugs are intended to target viral replication specifically; however, antivirals can have off-target effects including the inhibition of bacterial growth[9–12]. The antibacterial activity of several antiviral drugs has led to increased interest over the last decade in repurposing nucleoside analog antivirals, in particular, to treat multidrug resistant bacterial infections[13–15]. However, the antibacterial activity of many antiviral drugs also raises the question of whether antivirals can contribute to antimicrobial resistance, which encompasses the presence, development, dissemination, and treatment of antimicrobial resistant infections. Globally, antimicrobial resistance is one of the top ten threats to human health[16]. In the United States alone, there are over 2.8 million antibiotic-resistant infections every year and over 35,000 deaths due to antibiotic resistance[17]. Most discussion on antimicrobial resistance control centers on the judicious use of prescription antibiotics and hygiene[18,19]. However, an understanding of the full spectrum of contributors to antimicrobial resistance remains incomplete. Non-antibiotic pharmaceuticals such as antivirals may also contribute to the development of antimicrobial resistance on a global scale, but research is needed to gain full understanding of how drugs with antibacterial properties can lead to mutations in bacteria and cause cross-resistance to antibiotics.

While the antibacterial properties of many antiviral drugs have been demonstrated, it is largely unknown whether bacteria can become resistant to these antivirals and whether resistance mutations to antibacterial antivirals can confer cross-resistance to other antibacterial agents including antibiotic drugs. Non-pharmaceutical compounds with antimicrobial effects, such as some pesticides and herbicides, have been shown to contribute to antimicrobial resistance[20–22]. Other non-antibiotic drugs such as antidepressants and antiepileptic drugs have been shown to ultimately influence resistance by increasing rates of horizontal gene transfer, affecting the same or similar targets as antibiotic drugs, or resulting in changes to gene expression impacting resistance to antibiotic drugs[23–25]. Therefore, examining both the phenotypic and genotypic impacts of non-antibiotic drug exposure such as antivirals on bacteria can lead to better understanding of cross-resistance and the mechanisms leading to resistance. Beyond commonly known genetic mutations that cause antibiotic resistance, novel mutations due to non-antibiotic drug exposure may be responsible for conferring antibiotic cross-resistance[26]. By conducting whole genome sequencing, novel resistance mutations can be identified to help explain resistance phenotypes[27,28].

In this study, we investigated the antibacterial efficacy of antivirals and the potential for cross-resistance to other antivirals and antibiotic drugs. Using a culture-based approach with the model bacteria E. coli and B. cereus, we demonstrate that antiviral drugs have the potential to contribute to antibiotic resistance in gram-negative and gram-positive bacteria. While other studies have previously demonstrated the antibacterial properties of some antiviral drugs, to our knowledge, no other work has yet reported the extent to which numerous antivirals of different classes may lead to antibiotic cross-resistance. The results of this study suggest that antiviral drugs with antibacterial properties can impact the antimicrobial resistance phenotype of bacteria; further investigation may help elucidate the specific mechanisms by which these drugs and potentially others may act on bacteria and contribute to antibiotic cross-resistance.

## Results

**Antibacterial effects of antivirals on 24-h growth of E. coli and B. cereus.** Fourteen antiviral drugs from four different antiviral classes (antiherpetic, nucleoside reverse transcriptase inhibitor (NRTI), integrase inhibitor (II), non-nucleoside reverse transcriptase inhibitor (NNRTI)) with a range of different targets/modes of action (Supplementary Data 1) were screened for antibacterial activity against E. coli and B. cereus.

Demonstrating consistent results with previous work[10], eight of the fourteen antivirals tested inhibited the growth of E. coli, while only three of the fourteen displayed antibacterial activity against B. cereus (Fig. 1). The antivirals abacavir, darunavir, nevirapine, tenofovir, favipiravir, and emtricitabine had no significant impact on the growth of either E. coli or B. cereus up to 100 μg/mL over 24 h (Supplementary Fig. 1, Supplementary Fig. 2). Dolutegravir, efavirenz, and raltegravir significantly reduced the growth of both E. coli and B. cereus, while the nucleoside analogs acyclovir, didanosine, lamivudine, stavudine, and zidovudine significantly inhibited E. coli but not B. cereus (Fig. 1). Among the antivirals that displayed antibacterial activity, growth inhibition was typically observed at concentrations $\geq 50$ μg/mL; however, zidovudine significantly inhibited E. coli growth at 0.1 μg/mL (Fig. 1). Growth inhibition, as seen by a significant reduction in optical density (OD), was observed within 4 h of exposure for several of the antivirals with antibacterial activity against E. coli, and all antivirals with antibacterial activity inhibited growth within 8 h (Fig. 1). In some cases, the significance of antimicrobial effect was maintained over time, while in other cases it changed over time, either increasing or decreasing (Fig. 1). We were conservative with what we considered a significant inhibition of growth, and we required both significant ($p < 0.005$) and substantial (>10%) reduction in growth of the antiviral-treated bacteria compared to untreated controls (Supplementary Data 3, Supplementary Fig. 2). Differences were determined significant if the $p$-value of the $t$-test met the threshold ($p < 0.005$). All significant differences were further differentiated according to their degree of significance (from most to least significant: $p < 0.00005$, $p < 0.0005$, $p < 0.005$) for comparative purposes. Results that were not significant were interpreted as no significant difference in growth between the drug-treated condition and the untreated condition.

**Antiviral resistance and cross-resistance after repeated antiviral exposure.** After demonstrating that eight of the tested antivirals had significant antibacterial effects on the growth of E. coli and three impacted the growth of B. cereus, we explored the

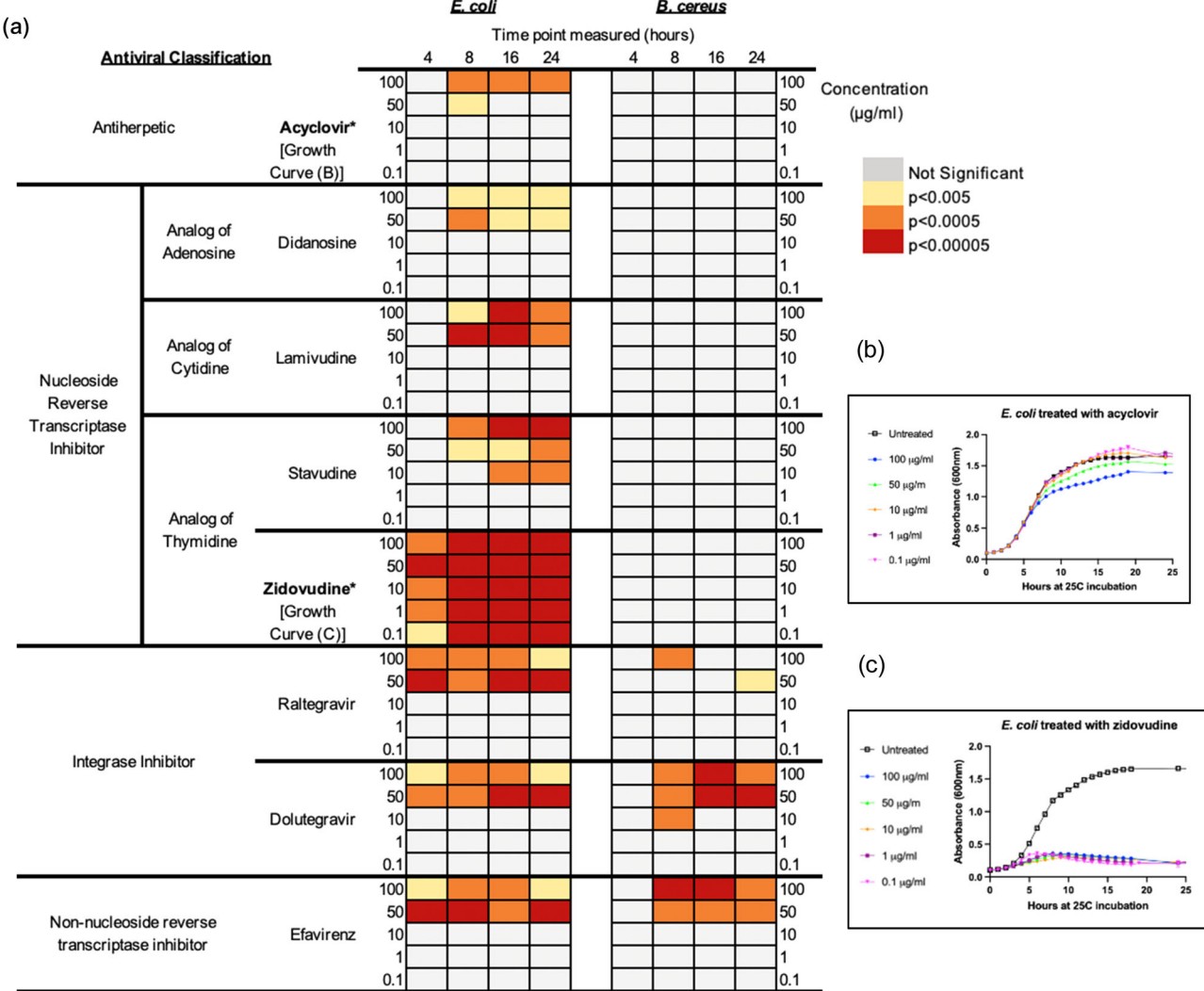

**Fig. 1 Antibacterial effects of antivirals on *E. coli* and *B. cereus*. a** Antibacterial effects of antivirals after 4–24 h of exposure at different antiviral concentrations ranging from 0.1–100 μg/mL. Nucleoside analogs acyclovir, didanosine, lamivudine, stavudine and zidovudine impact the growth of *E. coli* only. Both *E. coli* and *B. cereus* are significantly inhibited (with at least 10% reduction in growth compared to control) by non-nucleoside reverse transcriptase inhibitor efavirenz and the integrase inhibitors dolutegravir and raltegravir at concentrations from 10–100 μg/mL beginning at 4 h of exposure time for *E. coli* and 8 h for *B. cereus*. **b** Example of growth curves for *E. coli* treated with 0.1–100 μg/mL acyclovir over 24 h. The most significant difference in growth, as observed in the growth curve and noted in the heat map (**a**) is only from 8–24 h with 100 μg/mL acyclovir. **c** Example of growth curves for *E. coli* treated with 0.1–100 μg/mL zidovudine over 24 h. Significant differences between untreated and zidovudine-treated *E. coli* are observable at all concentrations tested and across all time points from 4–24 h.

potential for bacteria to become resistant to the antibacterial effects of these antivirals upon repeated exposure to the investigated drug. Antiviral-resistant strains that were isolated included zidovudine-resistant, lamivudine-resistant, stavudine-resistant, and didanosine-resistant *E. coli*. Dolutegravir-resistant and raltegravir-resistant strains were isolated for both *E. coli* and *B. cereus*. Resistant strains achieved similar maximum growth and growth kinetics compared to wild type, with the exception of the *E. coli* zidovudine-resistant strain which achieved 32% lower maximal growth compared to wild type (Supplementary Data 4; Supplementary Fig. 3). During the incubation period of 4–12 h, while bacteria were in log phase and OD was rapidly increasing, wild type *B. cereus* showed an average increase in OD of 0.162 (absorbance, 600 nm) per hour, and the antiviral-resistant strains showed an increase of 0.154 per hour. Wild-type *E. coli* showed an OD increase of 0.162 per hour, and antiviral-resistant strains (excluding zidovudine-resistant *E. coli*) showed an average increase in OD of 0.199 per hour (Supplementary Data 4).

Resistance to antivirals was observed at sub-inhibitory concentrations, typically at <100 μg/mL (Supplementary Data 5; Supplementary Fig. 4). Resistance was not observed in efavirenz treatments for both *E. coli* and *B. cereus*; resistance was observed in *E. coli* acyclovir treatments, but a resistant strain was not isolated (Supplementary Data 5).

Resistance to one antiviral drug conferred cross-resistance to other antiviral drugs for all six of the *E. coli* and both *B. cereus* resistant strains developed (Fig. 2, Supplementary Data 6). Although each strain displayed a unique pattern of cross-resistance to other antivirals, cross-resistance profiles clustered in some cases by the antiviral class from which each strain was isolated (Supplementary Fig. 5). For example, *E. coli* mutants isolated from the NRTI treatments lamivudine and stavudine were cross-resistant to the NRTIs lamivudine, stavudine and didanosine but susceptible to zidovudine (NRTI), dolutegravir (II), and efavirenz (NNRTI). These strains clustered together and were also similar to the NRTI didanosine-resistant strain (whose

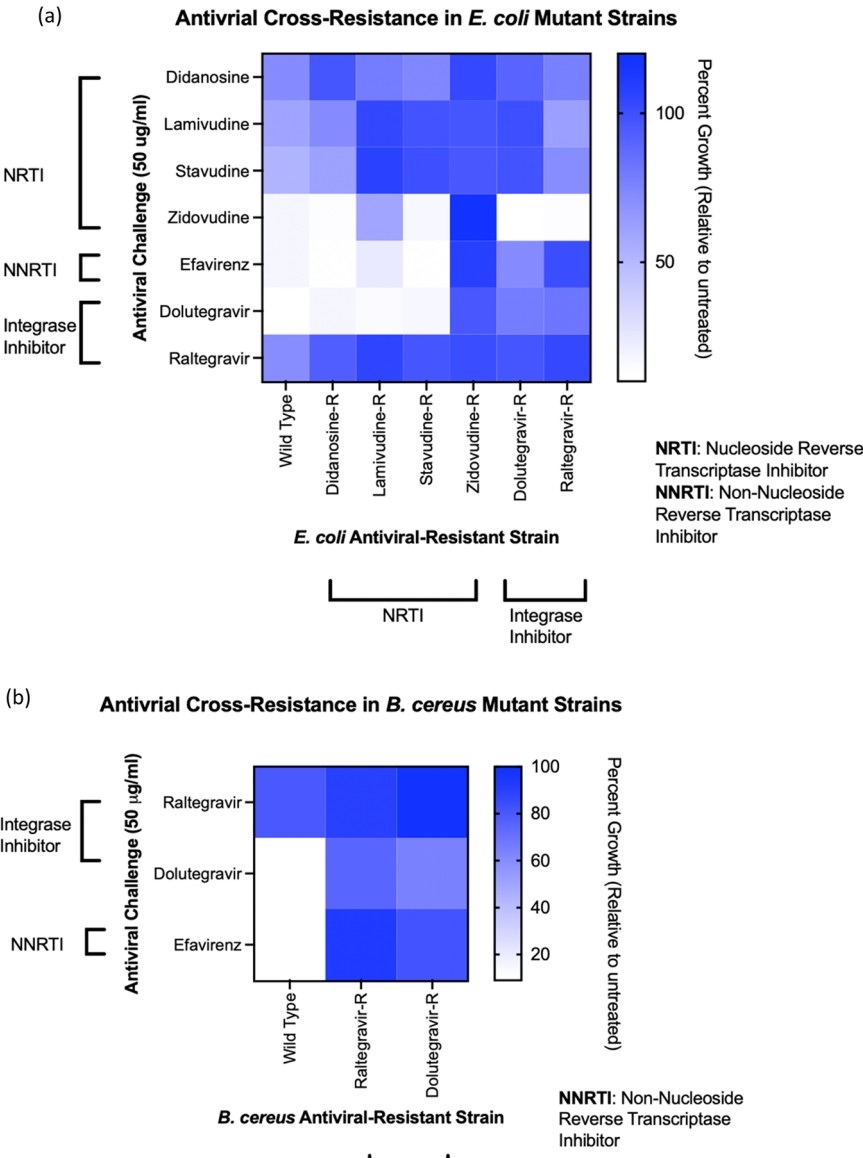

**Fig. 2 Antiviral-resistant *E. coli* and *B. cereus* exhibit cross-resistance to structurally and functionally distinct antiviral compounds. a** Compared to wild type *E. coli*, the antiviral-resistant strains show cross-resistance to multiple antivirals, and unique resistance profiles are observed for all *E. coli* mutant strains resistant to NRTIs (didanosine-resistant, Didanosine-R; lamivudine-resistant, Lamivudine-R; stavudine-resistant, Stavudine-R; zidovudine-resistant, Zidovudine-R) and integrase inhibitors (dolutegravir-resistant, Dolutegravir-R; raltegravir-resistant, Raltegravir-R). **b** Dolutegravir-R and Raltegravir-R *B. cereus* strains are resistant to efavirenz and dolutegravir compared to wild type *B. cereus*.

profile was more closely related to wild type *E. coli* rather than the lamivudine- and stavudine-resistant strains). Likewise, strains resistant to integrase inhibitors dolutegravir and raltegravir displayed similar cross-resistance profiles and clustered together. These strains were characterized by broader cross-resistance than the NRTI-resistant strains and were resistant to all tested antivirals except zidovudine. Despite being an NRTI like lamivudine, stavudine and didanosine, the zidovudine-resistant *E. coli* strain exhibited a uniquely broad cross-resistance profile that was more similar to the resistance profiles of the integrase inhibitor-resistant strains but stood out uniquely. Zidovudine-resistant *E. coli* was the only nucleoside-analog-resistant strain to show cross-resistance to NNRTI efavirenz compared to wild type (40% growth compared to 18% growth for wild type *E. coli*) as well as zidovudine itself.

Isolated *E. coli* and *B. cereus* strains resistant to integrase inhibitors dolutegravir and raltegravir demonstrated similarities in their cross-resistance to efavirenz, dolutegravir and raltegravir (Fig. 2a, b). Raltegravir-resistant *B. cereus* were resistant to efavirenz compared to wild type *B. cereus* (93% growth versus 9% growth). Interestingly, the raltegravir-resistant *B. cereus* mutant strain showed greater resistance to dolutegravir compared to the "dolutegravir-resistant strain" that was developed in the presence of dolutegravir and isolated for cross-resistance testing (Fig. 2b). Antiviral-resistant strains were, overall, shown to have stable resistance phenotypes, as demonstrated over 15 passages. For example, dolutegravir-resistant *B. cereus* maintained resistance to raltegravir, dolutegravir, and efavirenz throughout passaging in the absence of dolutegravir; zidovudine-resistant *E. coli* maintained resistance to zidovudine throughout 15 passages in absence of zidovudine (Supplementary Data 7).

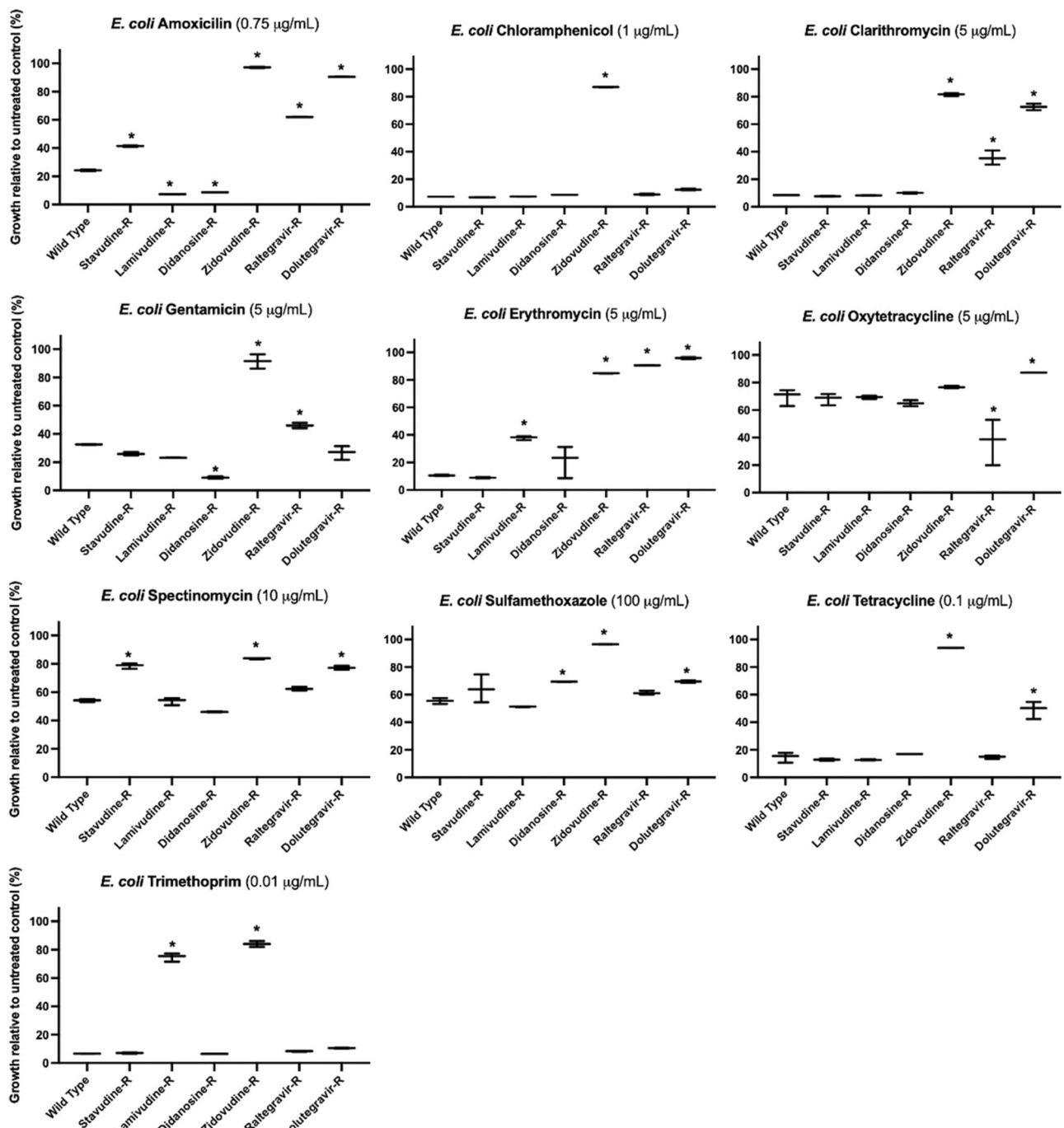

**Fig. 3 Antibiotic cross-resistance of antiviral-resistant *E. coli*. Antiviral-resistant *E. coli* are cross-resistant to antibiotics including clarithromycin and erythromycin, tetracycline and trimethoprim.** However, some antiviral-resistant *E. coli* mutants are more susceptible to some antibiotics compared to wild type *E. coli*, including increased susceptibility of didanosine-resistant (Didanosine-R) *E. coli* to amoxicillin and gentamicin. Asterisks (*) indicate significant difference compared to wild type ($p < 0.05$ and at least 10% substantial difference in growth between treated wild type and resistant strain); bars represent maximum, minimum and mean values for $n = 3$ absorbance measurements.

**Antibiotic cross-resistance in antiviral-resistant *E. coli* and *B. cereus*.** We further explored the extent of acquired resistance in *E. coli* and *B. cereus* by challenging the isolated antiviral-resistant strains with ten commonly prescribed broad-spectrum antibiotics (Supplementary Data 8). The final concentrations of antibiotics used to challenge wild type and mutant *E. coli* and *B. cereus* were selected after first challenging each individual strain and then choosing a concentration that captured the range of response to drug—i.e., changes in growth– across wild type and antiviral-

resistant mutant strains for comparative analysis. Similar to their cross-resistances with other antivirals, antiviral-resistant *E. coli* and *B. cereus* strains each exhibited unique resistance/susceptibility patterns to antibiotics (Fig. 3, Fig. 4, see Supplementary Data 9 for determinations of significance). Some mutant strains demonstrated strong resistance to antibiotics, and some strains were more susceptible to antibiotics compared to wild type control. *E. coli* and *B. cereus* mutants resistant to the same antiviral exhibited different antibiotic cross-resistance. For example,

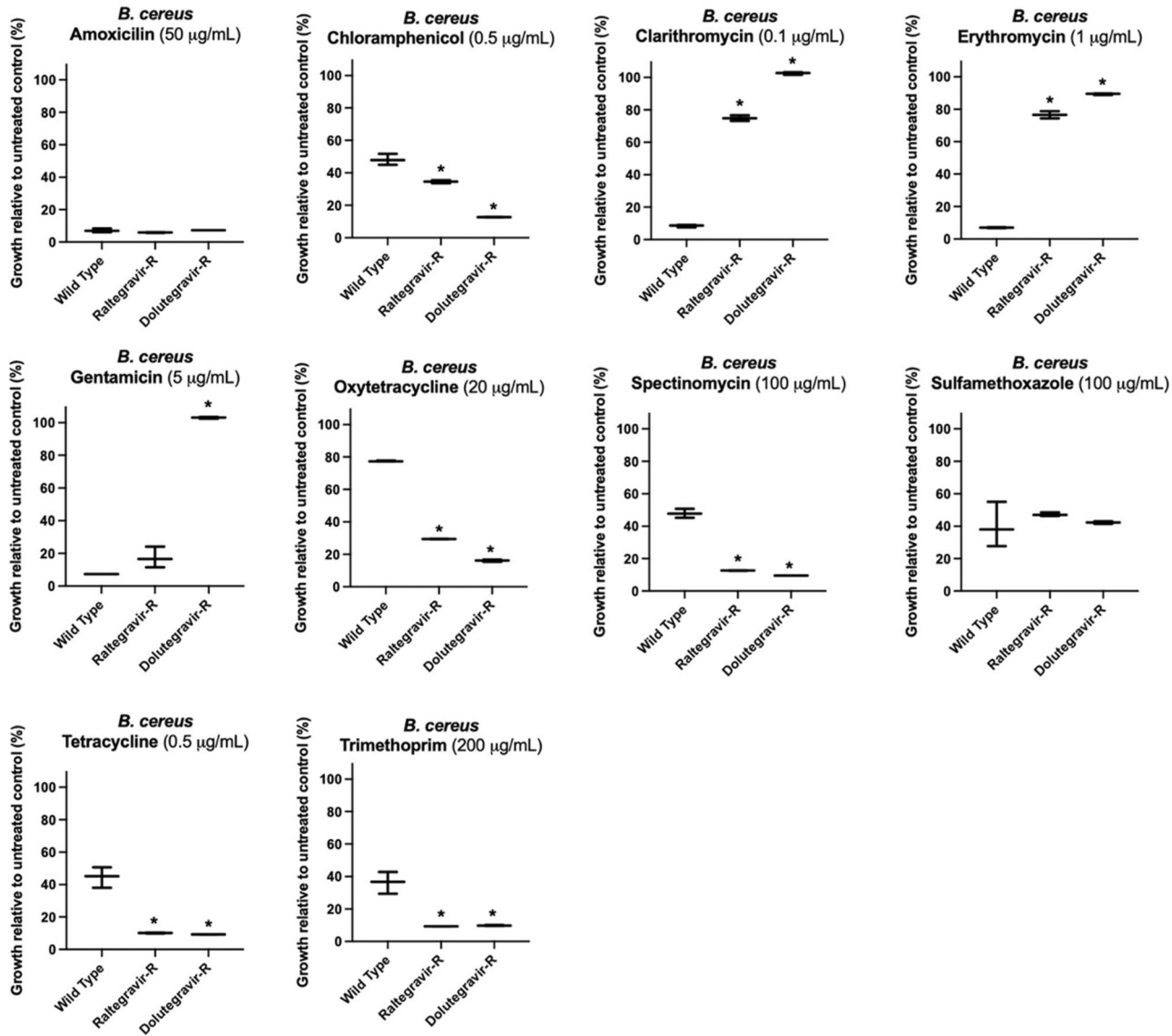

**Fig. 4 Antibiotic cross-resistance of antiviral-resistant *B. cereus*. Similar to antiviral-resistant *E. coli*, antiviral-resistant *B. cereus* are cross-resistant to the macrolide antibiotics clarithromycin and erythromycin.** However, *B. cereus* antiviral-resistant mutants raltegravir-resistant (Raltegravir-R) and dolutegravir-resistant (Dolutegravir-R) were often equally or more susceptible to antibiotics compared to wild type *B. cereus*, as illustrated by their response to oxytetracycline or spectinomycin. Asterisks (*) indicate significant difference compared to wild type ($p < 0.05$ and at least 10% substantial difference in growth between treated wild type and resistant strain); bars represent maximum, minimum and mean values for $n = 3$ absorbance measurements.

dolutegravir-resistant *E. coli* showed some resistance to tetracycline and oxytetracycline (Fig. 3) while dolutegravir-resistant *B. cereus* were more susceptible to tetracycline and oxytetracycline compared to wild type (Fig. 4).

Most antiviral-resistant *E. coli* mutant strains exhibited several strong antibiotic cross-resistances. Raltegravir-resistant, dolutegravir-resistant, and zidovudine-resistant *E. coli* showed resistance to clarithromycin and erythromycin (macrolide antibiotics) (Fig. 3). Zidovudine-resistant and lamivudine-resistant *E. coli* were resistant to trimethoprim. Dolutegravir-resistant and zidovudine-resistant *E. coli* strains showed cross-resistance to tetracycline. Compared to wild-type *E. coli*, dolutegravir-resistant *E. coli* also showed decreased susceptibility to amoxicillin, spectinomycin, and sulfamethoxazole (24 vs 90%, 54 vs 77%, 56 vs 70%, wild type vs dolutegravir-resistant growth, respectively) in addition to the macrolide and tetracycline antibiotics (Fig. 3, Supplementary Data 10). Some antiviral-resistant *E. coli* strains

were more susceptible to antibiotics compared to wild type *E. coli*. Greater susceptibility was observed for lamivudine-resistant and didanosine-resistant *E. coli* to amoxicillin and raltegravir-resistant *E. coli* to oxytetracycline. When treated with gentamicin (aminoglycoside), both raltegravir-resistant and zidovudine-resistant *E. coli* were more resistant compared to wild type *E. coli* (46% vs 32% growth for raltegravir-resistant, and 92% vs 32% for zidovudine-resistant *E. coli*).

*B. cereus* raltegravir-resistant and dolutegravir-resistant strains were either more susceptible or comparably susceptible to seven out of ten antibiotics tested compared to the wild type *B. cereus*. Both dolutegravir-resistant and raltegravir-resistant *B. cereus* were equally susceptible to amoxicillin and sulfamethoxazole compared to wild type (Fig. 4). The strains were more susceptible to spectinomycin, tetracycline, trimethoprim, oxytetracycline and chloramphenicol compared to wild type *B. cereus* (Fig. 4). However, for the antibiotics gentamicin, erythromycin and

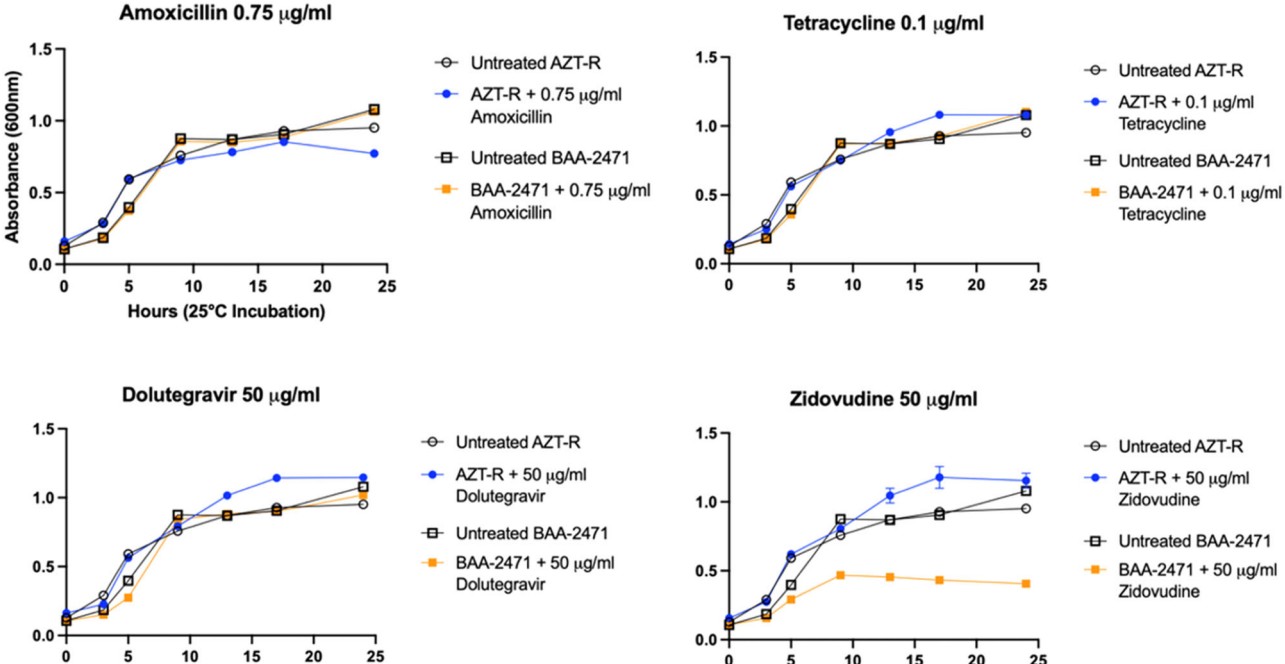

**Fig. 5 Comparison of BAA-2471 and zidovudine-resistant *E. coli* growth kinetics and resistance.** As expected, multidrug-resistant clinical *E. coli* isolate BAA-2471 is resistant to amoxicillin and tetracycline at the concentrations tested, while wild type *E. coli* were susceptible at these concentrations (wild type response shown in Fig. 3). The growth curves of zidovudine-resistant *E. coli* (AZT-R) demonstrate the same resistance as BAA-2471 to the antibiotics as well as the antiviral dolutegravir. However, BAA-2471 is susceptible to zidovudine. For each time point, $n = 3$ technical replicates; mean absorbance is plotted for each time point, and absence of observable error bars at any data point is indicative of close alignment of triplicate measurements.

clarithromycin, the antiviral-resistant strains showed greater resistance compared to wild type *B. cereus*: only dolutegravir-resistant *B. cereus* was resistant to gentamicin; similar to dolutegravir-resistant and raltegravir-resistant *E. coli*, dolutegravir-resistant and raltegravir-resistant *B. cereus* both were resistant to clarithromycin and erythromycin. When treated with 1 µg/mL erythromycin, wild type *B. cereus* exhibited on average only 7% growth relative to untreated wild type, while dolutegravir-resistant *B. cereus* exhibited 90% growth relative to untreated dolutegravir-resistant. Raltegravir-resistant *B. cereus* exhibited 76% growth relative to untreated raltegravir-resistant in the presence of erythromycin. When exposed to 0.1 µg/ml clarithromycin, wild type *B. cereus* exhibited 9% growth while dolutegravir-resistant and raltegravir-resistant strains exhibited 103% growth and 75% growth, respectively.

**Comparative phenotypic analysis of multidrug-resistant *E. coli*.** Following our results demonstrating that antiviral-resistant *E. coli* can exhibit broad cross-resistance to antibiotic drugs, we wanted to contextualize the clinical relevance of the survival and growth of antiviral-resistant *E. coli*. To validate the antibiotic cross-resistance phenotypes of the antiviral-resistant strains and the in vitro method of evaluation for resistance, we challenged a well-characterized multidrug-resistant *E. coli* strain (ATCC BAA-2471, clinical respiratory sample) with the same antibiotics and concentrations used to challenge antiviral-resistant *E. coli* strains. *E. coli* BAA-2471 exhibits resistance to many classes of antibiotics including penicillins, cephalosporins, carbapenems, quinolones, aminoglycosides, and others[29]. The strain is therefore resistant to antibiotics we tested including amoxicillin, gentamicin, tetracycline, trimethoprim and sulfamethoxazole. We confirmed the resistance of BAA-2471 to these antibiotics using our methods, and we also compared the strain's resistance or susceptibility to antiviral drugs (Fig. 5, Supplementary Data 11). Zidovudine-

resistant *E. coli*—the antiviral-resistant *E. coli* strain with the broadest resistance profile—and untreated BAA-2471 exhibited comparable growth kinetics. Resistance was indeed observed for all antibiotics to which BAA-2471 was purportedly resistant, and the growth of BAA-2471 and zidovudine-resistant *E. coli* in the presence of antibiotic treatment was comparable (Fig. 5, Supplementary Data 15). While both BAA-2471 and zidovudine-resistant *E. coli* demonstrated growth in the presence of 3 µg/ml gentamicin, we found growth of BAA-2471 inhibited at 16 h' incubation with 5 µg/ml gentamicin in our assays whereas zidovudine-resistant *E. coli* were not susceptible (Fig. 3, Supplementary Data 11). In the presence of 50 µg/ml zidovudine, after 16 h' incubation, zidovudine-resistant *E. coli* exhibited 120% growth compared to untreated control whereas BAA-2471 exhibited only 48% growth (Fig. 5, Supplementary Data 15). These results both validated our approach to assessing the phenotypic resistance profiles of resistant strains and allowed comparative analysis of bacterial growth in the presence and absence of antiviral drugs and antibiotics using a well-characterized resistant clinical isolate.

Further, the results suggest that zidovudine-resistant *E. coli* exhibit resistance mutations that are unique compared to the clinical isolate BAA-2471 yet grant comparable fitness in the absence of drug and comparable survival in the presence of antibiotic concentrations that are inhibitory for wild type.

**Comparative genome analysis of antiviral-resistant *E. coli*.** After observing that antiviral-resistant isolates of *E. coli* and *B. cereus* have phenotypically unique changes in resistance to antibiotics compared to wild type, we conducted whole genome sequencing of antiviral-resistant *E. coli* strains with the goal of identifying genetic changes that may explain resistance to either antiviral drugs or antibiotics. Comparative genomics of *B. cereus* isolates could not be conducted due to contamination during

| Antiviral-Resistant E. coli Genome[+] | Mutation* | Gene | Function/Characteristics | Potential Role in Antimicrobial Resistance | Percent Protein Truncated Due to Mutation |
|---|---|---|---|---|---|
| Didanosine-resistant [CP115968] | (232) L: CTG → R: CGA | ABC transporter permease yhdY | • Transmembrane transport of substituents including carbohydrates, lipids, proteins, drugs[51] | • ABC transporter family involved in drug efflux[43] | 35.2 |
| | (47) S: AGGT → R: AGG (48) E: GAG → R: AGG (50) E-K: GAA-AAA → R: CGA (53) G: GGA → R: CGC | Galactofuranose-binding protein ytfQ | • Galactofuranose transport during carbon scavenging conditions • Possible additional functions[52] | • Certain SNPs predictive of resistance to subinhibitory concentrations of antibiotics[50] | 83.4 |
| Lamivudine-resistant [CP115179] | (108) C: TGT → V: GTA | Hexose-6-phosphate:phosphate antiporter uhpT | • Transporter for phosphate and glucose-, fructose- or mannose-6-phosphate[57] | • Role in antibiotic resistance[47] | 73.1 |
| | (27) E: GAA → (non-coding region) | Bifunctional ligase/repressor birA | • Bifunctional biotin ligase and biotin operon repressor[58] | • Proposed target to treat multi-drug resistant infection[59,60] | 90.5 |
| Stavudine-resistant [CP115180] | (75) V-E: GTT-GAA → V-N: GTG-AAC | tRNA$_1^{Val}$ (adenine$^{37}$-N$^6$)-methyltransferase yfiC | • Methylates adenine in position 37 of tRNA$_1^{Val}$[61] | • May be implicated in antimicrobial resistance[48,49] | 52.0 |

**Fig. 6 Genetic changes in the didanosine-resistant, lamivudine-resistant, stavudine-resistant E. coli genomes with known or suggestive roles in antimicrobial resistance.** Blue highlighting indicates known role in antimicrobial resistance; green highlighting indicates recently suggested role in resistance. [+]Accession number listed below each genome. *"Mutation" column lists first in parentheses the codon number in the wild type reference gene that is mutated compared to the antiviral-resistant strain, followed by the mutation described as the single letter amino acid abbreviation and sequence of the wild type and the single letter amino acid abbreviation and altered sequence in the antiviral-resistant mutant gene.

reference strain library preparation and small number of mutant strains. Overall, there were as few as two and as many as 23 unique genetic changes in the antiviral-resistant E. coli mutant genomes as compared with the reference strain that could not be attributed to either a change in the reference during sequencing (differences found in all mutant genomes) or to a sequencing error (e.g., insertion of A within A homopolymer sequence) (Supplementary Data 13). Mutations observed in three of the mutant strains (raltegravir-resistant E. coli, [CP117043]; dolutegravir-resistant E. coli, [CP117044]; didanosine-resistant E. coli, [CP115968]; Fig. 6, Fig. 7) had genetic changes in genes with established roles in antibiotic resistance. Raltegravir-resistant and dolutegravir-resistant E. coli isolates—both integrase inhibitor resistant strains– have the same base change (A→G) leading to a non-synonymous amino acid change (Y→C) in the cAMP-activated global transcriptional regulator crp (Fig. 7), associated with resistance to macrolides, potentially explaining part of their acquired antibiotic resistance phenotype of these two antivirals from the same antiviral class. A mutation in the multidrug efflux pump, efflux resistance-nodulation-division (RND) transporter permease acrB, was observed in the dolutegravir-resistant E. coli isolate. In this case, the observed mutation led to a 1.8% increase in the length of the protein (Fig. 7) and may explain the observed resistance to tetracycline[30–33]. In didanosine-resistant E. coli, a deletion was observed in ABC transporter permease yhdY (single base pair deletion in codon 232), suggesting potentially altered function to this protein family involved in the transport of a variety of substituents including not only carbohydrates, proteins, and lipids but also pharmaceutical drugs[34] (Fig. 6). The early stop codon introduced by the single base pair deletion led to a 35.2% reduction in protein length. Additionally, two mutants exposed to nucleoside analogs (lamivudine-resistant E, coli, [CP115179], cytidine analog; zidovudine-resistant E, coli, [CP117469],

thymidine analog) had different genetic changes within thymidine kinase gene (tdk), suggesting a role in nucleoside analog antiviral drug resistance (Supplementary Data 13).

Other mutations causing substantial changes in protein sequences observed in antiviral-resistant E. coli were located in genes implicated in antibiotic resistance in recent studies. For example, mutations were observed in the coding region for the phosphotransferase system (PTS) N-acetylgalactosamine-specific transporter subunit IIB (agaV) in dolutegravir-resistant E. coli. This system is involved in carbohydrate transport, yet mutations in PTS have also been associated with pan-tolerance to antimicrobials[26] (Fig. 7). The mutation in agaV led to 47.6% reduction in the length of the protein. Also, in dolutegravir-resistant E. coli, a five-base pair insertion was observed in the sensory domain of two-component sensor kinase family protein, which consists of a histidine kinase and cognate response regulator. Various mutations in this family of proteins have been linked to antimicrobial resistance[35]. Lamivudine-resistant E. coli exhibited a single base pair deletion in the region coding for hexose-6-phosphate:phosphate antiporter (uhpT), leading to 73.1% truncation of the protein (Fig. 6). This antiporter is involved in the exchange of hexose-6-phosphate (which could include the six-carbon sugars glucose-6-phosphate, fructose-6-phosphate and mannose-6-phosphate) and phosphate across the cell membrane, and it has also been implicated in antibiotic resistance[36]. A single base pair deletion was observed in tRNA$_1^{Val}$ (adenine$^{37}$-N$^6$)-methyltransferase in stavudine-resistant E. coli leading to truncation of 52% of the protein (Fig. 6). This particular methyltransferase methylates adenine in position 37 of tRNA$_1$ (Val). Didanosine-resistant E. coli also exhibited four single base pair deletions in codon 50 of galactofuranose-binding protein (ytfQ), an ABC transporter involved in galactofuranose transport typically in carbon scavenging conditions but recently as a predictive gene for antimicrobial resistance[37] (Fig. 6). The

| Antiviral-Resistant *E. coli* Genome[+] | Mutation* | Gene | Function/Characteristics | Potential Role in Antimicrobial Resistance | Percent Protein Truncated Due to Mutation |
|---|---|---|---|---|---|
| Dolutegravir-resistant [CP117044] | (non-coding region) → (1) M: ATG | Resistance-nodulation-division (RND) transporter permease *acrB* | • Multidrug efflux pump[39–42] | • Mechanism for drug resistance[39–42] | -1.8[#] |
| | (19) Y: TAC → C: TGC | cAMP-activated global transcriptional regulator *crp* | • Global transcriptional regulator[26,53] | • Repression of the genes encoding the MdtEF multidrug efflux pump[54] | 0 |
| | (77) S: TCA → STOP TAA | Phosphotransferase system (PTS) N-acetylgalactosamine-specific transporter subunit IIB *agaV* | • Carbohydrate transport[55,56] | • Associated with pan-tolerance to antimicrobials[26] | 47.6 |
| | (292) A-E: GCC-GAA → A-A: GCC-GCG | Sensory domain of two-component sensor histidine kinase *dcuS* | • Directs response to variety of environmental stimuli[46] | • May be involved in antibiotic resistance[46] | 44.1 |
| Raltegravir-resistant [CP117043] | (19) Y: TAC → C: TGC | cAMP-activated global transcriptional regulator *crp* | • Global transcriptional regulator[26,53] | • Repression of the genes encoding the MdtEF multidrug efflux pump[54] | 0 |
| Zidovudine-resistant [CP117469] | (54) T-I: ACC-ATT → T-L: ACA-TTA | Bifunctional aspartate kinase/homoserine dehydrogenase I *thrA* | • Involved in biosynthesis of amino acids, nucleotides[62] | • Potential target for multi-drug-resistant infection[63] | 90.9 |

**Fig. 7 Genetic changes in the dolutegravir-resistant, raltegravir-resistant, zidovudine-resistant *E. coli* genomes with known or suggestive roles in antimicrobial resistance.** Blue highlighting indicates known role in antimicrobial resistance; green highlighting indicates recently suggested role in resistance. [+]Accession number listed below each genome. *"Mutation" column lists first in parentheses the codon number in the wild type reference gene that is mutated compared to the antiviral-resistant strain, followed by the mutation described as the single letter amino acid abbreviation and sequence of the wild type and the single letter amino acid abbreviation and altered sequence in the antiviral-resistant mutant gene. [#]The protein increased in length by 1.8% due to the observed mutation.

sodium/glutamate symporter possessed a single base pair insertion leading to the truncation of ten percent of the protein in dolutegravir-resistant *E. coli*. Although more work is needed to determine which of the genetic changes observed in the mutant genomes are important for the acquired antiviral or antimicrobial phenotypes, we have presented several likely candidates with a known role in antibiotic resistance, and which have led to the largest changes in protein sequences.

To investigate whether the mutations potentially implicated in antiviral resistance and antibiotic cross-resistance could be observed in a characterized multidrug-resistant *E. coli* model, we examined the publicly available full genome sequence for BAA-2471. When comparing multidrug-resistant *E. coli* BAA-2471 to antiviral-resistant *E. coli* strains, none of the mutations observed in antiviral-resistant *E. coli* strains leading to protein truncation were observed in BAA-2471 (Supplementary Data 14). For example, the mutations in thymidine kinase and bifunctional aspartate kinase/homoserine dehydrogenase that characterized zidovudine-resistant *E. coli* were not observed in BAA-2471. All the genes that contained insertions or deletions in antiviral-resistant *E. coli* mutants were intact for BAA-2471 and comparable to wild type *E. coli*.

## Discussion

Despite the increasing public health concerns regarding the development and spread of antimicrobial resistance, the agents contributing to resistance development beyond antibiotic drugs remain incompletely understood[21,38–42]. Efforts to control antibiotic resistance are primarily focused on antibiotic stewardship for the prescription of antibiotic drugs[43] but without attention to other drugs with antimicrobial properties. Antivirals have global widespread use, known antibacterial effects, and have even been considered as treatment for drug-resistant bacterial infections[13,15]. To date, no studies we are aware of have explored the potential for antiviral drugs to contribute to antimicrobial resistance. In this work, we examined the antibacterial activity of

fourteen antiviral drugs of different antiviral classes to determine their potential to contribute to antiviral resistance and antibiotic cross-resistance. Our findings indicate that in addition to antibiotic drugs, antiviral drugs that exhibit antibacterial properties may also contribute to the antibiotic resistance burden.

Gram-negative *E. coli* and gram-positive *B. cereus* were chosen as well-characterized model organisms for investigation with environmental and clinical relevance[44–46]. Antivirals were tested at concentrations ranging from 0.1–100 µg/mL over 0–24 h (Fig. 1). The concentration range tested covers inhibitory and sub-inhibitory concentrations demonstrated in previous in vitro tests with bacteria[10–12]. We included concentrations tested in existing literature to ensure reproducible results and to validate our methods. The concentration range tested also includes therapeutic or circulating concentrations of antiviral drugs, such as the plasma concentration of zidovudine (0.016–1.7 mg/L)[47] and therapeutic window of efavirenz (1–4 mg/L)[48], as well as concentrations of antivirals detected in the aquatic environment that can reach concentrations in the µg/L range[49]. However, in attempt to establish inhibitory values for various incubation periods and exposure concentrations, the concentration ranges tested generally exceeded environmentally detected levels or a typical plasma concentration in the human body[50].

We successfully isolated antiviral-resistant *E. coli* and *B. cereus* mutants after repeatedly exposing the bacteria to antiviral drugs with demonstrated antibacterial properties. When antiviral-resistant strains were challenged with antibiotic drugs and other antiviral drugs, each resistant strain exhibited a unique phenotypic profile of resistance to antiviral and antibiotic drugs. The resistance and cross-resistance profiles of the antiviral-resistant strains did seem to cluster by antiviral drug class, but none exhibited identical phenotypic resistance. The results suggest that despite structural similarity based on drug class in some cases, the antiviral drugs may be acting through different and possibly multiple pathways to exert their antibacterial properties. Resistant mutants therefore may possess a unique set of mutations conferring resistance to antiviral drugs and antibiotics.

In some cases, antiviral-resistance mutations conferred increased susceptibility to some antibiotics when resistant strains were challenged to test for cross-resistance. However, the increased sensitivity to particular classes of antibiotics in the dolutegravir- and raltegravir-resistant *B. cereus* strains, for example, may not be surprising when considered alongside the results of other studies that have observed increased susceptibility of certain drug-resistant strains of bacteria to other classes of antibiotic drugs[28,51]. Fitness costs and other changes in susceptibility due to resistance mutations have been described elsewhere as well[39,52,53]. The reduced maximal growth of zidovudine-resistant *E. coli* compared to wild type (Supplementary Data 4) and comparable growth to multidrug-resistant BAA-2471 (Fig. 5) may suggest how the mutations observed in this resistant strain come at a metabolic cost to the microorganism. However, the stability of the mutations throughout passaging even in the absence of drug suggest the mutation is favorable to sustain, or difficult to revert[54] (Supplementary Data 7). Further investigation may help illuminate where and how the antiviral drugs exert their antimicrobial influence in cell replication or metabolism and how growth conditions also impact the effects of antimicrobials on bacteria[55]. The significance level of antimicrobial effect may help indicate that some drugs act early in cell killing while others only take effect after more extended incubation.

Whole genome sequencing results provided further evidence that each antiviral-resistant strain of *E. coli* possessed unique mutations conferring cross-resistance to different classes of antiviral drugs or antibiotics. Mutations primarily consisted of single base pair insertions and deletions. While the whole genome sequencing results obtained cannot directly explain the phenotypic resistance profiles without further in-depth biological investigation, the results hint at pathways and targets that may be involved in resistance caused by exposure to antiviral drugs. Two out of the six *E. coli* isolates with a mutation in the same gene (thymidine kinase gene *tdk*) suggest that altered nucleoside metabolism could be an important mechanism to protect against the antimicrobial effects of nucleoside analogs. Thymidine kinase phosphorylates thymidine and deoxyuridine, both of which are involved in numerous metabolic pathways in *E. coli*[56,57].

Two out of the six *E. coli* isolates also had the same genetic change in a global transcriptional regulator, *crp* also suggesting that gene regulation may play an important role. Also of note was the observed mutation in tRNA$_1^{Val}$ (adenine$^{37}$-N$^6$)-methyltransferase in stavudine-resistant *E. coli*. Recent studies have pointed to the role of tRNA modification genes in resistance to numerous classes of antibiotics[58,59]. Finally, mutations in several transporters were identified—including the ABC transporter permease *yhdY*, the PTS N-acetylgalactosamine-specific transporter *agaV*, the hexose-6-phosphate:phosphate antiporter *uhpT*, and ABC transporter galactofuranose-binding protein *ytfQ*—adding to the important role of membrane processes in antibiotic resistance mechanisms.

Although several of the observed genetic mutations in antiviral-resistant isolates occurred in proteins or pathways with known or suspected roles in antimicrobial resistance, the majority of genetic changes occurred in genes that are not known to be involved in antibiotic resistance. Interestingly, many of the genes in which the mutations occurred are known to be involved in aspects of metabolism and nutrient transport such as galactofuranose-binding protein, sodium/glutamate transporter, and bifunctional aspartate kinase/homoserine dehydrogenase I. Consistent with the existing literature that suggest an array of metabolic pathways may be implicated in antimicrobial resistance or may provide targets for novel therapies for multidrug-resistant bacterial infections[26,35,60–62], these mutations may still point to key genes involved in resistance to antiviral drugs and associated antibiotic cross-resistance. Many genes beyond those traditionally thought to be responsible for antibiotic resistance have been highlighted as relevant for clinical isolates with antibiotic resistance, such as mutations in N-acetylglycosaime deacetylase[63], glycerol kinase[27], glycosytransferase[64], serine-threonine kinase[65], and histidine kinase[66]. Given the interest in identifying mutations that could be causative of the resistance phenotype, mutations in genes such as those identified in this study may indicate how non-antibiotic drugs may still have antimicrobial effects and from where broad antimicrobial resistance may arise. In addition, the genes identified in this study may point to pathways that may be involved in resistance and therefore serve as targets for novel therapeutics beyond traditional antibiotics. The mutations identified in antiviral-resistant genomes may help explain antibiotic cross-resistance phenotypes due to antiviral exposure. The identified mutations may also provide further support that mutations in genes beyond traditional resistance genes may be implicated in a range of antimicrobial resistance.

Antiviral-resistant *E. coli* exhibited phenotypic resistance across many classes of structurally distinct antimicrobials, yet the whole genome sequencing results of antiviral-resistant *E. coli* revealed relatively few base pair mutations compared to wild type *E. coli*. Therefore, these few select mutations may have wide-ranging implications for broad cross-resistance. Most antiviral-resistant strains exhibited less than ten unique genetic changes compared to wild type. For instance, zidovudine-resistant *E. coli* by far exhibited the broadest phenotypic resistance profile to all antiviral and antibiotic drugs tested but exhibited only two unique genomic mutations in coding regions: in the thymidine kinase gene and in bifunctional aspartate kinase/homoserine dehydrogenase I. Other studies have associated a zidovudine resistant phenotype with mutations in thymidine kinase[9,67,68]. However, Doléans-Jordheim et al. assessed only spontaneous zidovudine resistant mutants—rather than mutants isolated after repeated zidovudine exposure— and genetic analysis was limited to only the thymidine kinase gene, rather than the whole genome. No evidence of antibiotic cross-resistance in spontaneous zidovudine-resistant mutants was reported by Doléans-Jordheim et al., but the stability of zidovudine resistance was observed, as it was in our study. It is possible that genetic mutation beyond thymidine kinase alone is responsible for the broad cross-resistance observed in our zidovudine-resistant *E. coli* and that repeated exposure to the drug is also necessary to induce resistance mutations that confer cross-resistance to other antimicrobials. No studies to our knowledge have investigated how mutations in bifunctional aspartate kinase/homoserine dehydrogenase I coupled with mutations in thymidine kinase confer cross-resistance or tolerance to multiple classes of antibiotics. Both thymidine kinase and bifunctional aspartate kinase/homoserine dehydrogenase I are ultimately involved in nucleotide metabolism[69] and specifically the pyrimidine thymidine. Homoserine dehydrogenase has recently been proposed as a target in systemic fungal infection[70]. Since zidovudine is an analog of thymidine, it is possible to hypothesize that a target of zidovudine may be the pathways involved in thymidine metabolism, which may also influence *E. coli*'s resistance to a spectrum of antibiotic drugs. Trimethoprim and sulfamethoxazole, specifically, target tetrahydrofolate metabolism, which also implicates thymidine and thymidine kinase[71]. In other contexts, zidovudine has been shown to prevent the transfer of antibiotic resistance genes by horizontal gene transfer[72]. Our studies provide evidence that chromosomal mutations that confer cross-resistance to the antiviral zidovudine and naïve cross-resistance to antibiotics can develop de novo due to the presence of the exposure drug. The ability of bacteria to adapt to environmental stressors including drugs may lead to many different alterations and adaptations for

survival[73,74]. Alterations in gene function can be both mechanisms to survive an environmental challenge and lead to or be associated with antimicrobial resistance[66,75,76]. Antiviral drugs may have similar effects as stressors leading to resistance phenotypes, although further study is necessary for establishing the pathways and mechanisms by which antiviral drugs affect bacteria.

Beyond oral antiviral therapy exposure and the accompanying risk of developing antimicrobial resistance in the human body, the presence of antivirals and antiviral-resistant bacteria from human waste in the aquatic environment poses a risk for the development and spread of antimicrobial resistance in the aquatic environment. Parallel environmental implications can be drawn between antibiotic drugs and antiviral drugs to underline the risks of antiviral drugs present in the environment. Antibiotic drugs such as amoxicillin, erythromycin, trimethoprim and sulfamethoxazole are commonly detected in surface waters and have been associated with the presence of resistance genes[77–81]. Antivirals with antibacterial properties have also been detected in the aquatic environment—from wastewater to surface water and drinking water[82]—but their contribution to the development and spread of resistance genes is unknown. Current wastewater treatment processes often do not sufficiently remove many antivirals from wastewater. Many antivirals persist through wastewater treatment[83–86] and are released into the environment, detectable at concentrations ranging from ng/L to μg/L. Antivirals with known antibacterial properties (i.e., zidovudine and efavirenz) have been detected in soils, surface water and drinking water in addition to wastewater[50,82,85,87,88]. The documented presence, persistence, and chronic low-level concentrations of antivirals in the environment is a similar scenario to the presence of antibiotics in the aquatic environment. Just as similarly to antibiotic drugs, the presence of antibacterial antivirals in the environment could exacerbate the antimicrobial resistance crisis as antiviral-cross-resistance genes develop and spread. The evidence of antivirals in the environment highlights the risk of bacterial and human exposure to antivirals beyond the context of medicative consumption.

In developing and isolating novel antiviral-resistant mutant strains of *E. coli* and *B. cereus*, we demonstrated how non-antibiotic drugs such as antivirals could contribute to antimicrobial resistance. Bacteria exposed to individual antiviral drugs developed stable mutations that conferred cross-resistance to other antibacterial antivirals and antibiotics. In the future, isolating and challenging a large number (>5) of antiviral-resistant isolates from each antiviral exposure will help answer whether antiviral-resistance mutations are consistent or stochastic. Given the diversity of gram-positive and gram-negative bacteria, using additional strains of *E. coli* and expanding this work beyond *E. coli* and *B. cereus* will help elucidate the broader implications for the effects of antibacterial antivirals on bacterial communities such as those found in environmental waters, wastewater treatment plants, and the human gut. Mechanisms of resistance to antiviral drugs are likely to vary across species and even strains of bacteria, and the patterns of cross-resistance that arise may therefore also be heterogeneous. The effects of mixtures of drugs—similar to those relevant in the human body or the highly heterogeneous context of wastewater—on bacteria also warrants further investigation. As the use and manufacture of antivirals continues to expand worldwide, it will be essential to realize the full potential contribution of antivirals to antimicrobial resistance both in the human body and in the environment. The risk of developing antimicrobial resistance due to the presence of antivirals may also be highest where the burden of viral disease— and concurrent use of antiviral drugs– is also highest. Lack of wastewater infrastructure is likely to compound the risk as untreated wastewater is discharged into the environment, and with it, high concentrations of antiviral drugs and bacteria.

## Methods

**Reagents**. All antiviral compounds were purchased from Toronto Research Chemicals and were certified >98% purity. Based on solubility, stock solutions of 1, 2 or 10 mg/mL were made in MilliQ water or DMSO and stored in amber vials in 4 °C. *E. coli* (B strain, product number 12-4300) was purchased from Carolina. *B. cereus* was purchased from Ward's Science (WARD470176-602). Multidrug-resistant *E. coli* (product number BAA-2471) was purchased from ATCC.

**Culture technique**. Bacteria were initially grown from agar slants in autoclaved media: LB media, Lennox (Fisher, BP1427-500, Lot 188454) for *E. coli* B or tryptic soy broth (TSB) media (Teknova, TO420, Lot T042010A2001) for *B. cereus*. Media were prepared according to manufacturer's instructions. After growing bacteria to stationary phase in sterile vented T25 flasks, bacteria were plated on agar plates and single colonies were selected to prepare glycerol stocks (50% glycerol, 50% bacteria in culture medium) for future use throughout experiments. To develop resistant strains, bacteria grown in flasks were incubated with 10–100 μg/mL antiviral overnight in 37 °C. Bacteria were collected by centrifuging in sterile microcentrifuge tubes for 1 min at 4000 RPM. Bacteria were washed with LB or TSB media, centrifuged a second time, then resuspended in media and added to a fresh flask with a total of 5 mL fresh media and the same concentration of antiviral previously. Bacteria were re-grown overnight in 37 °C, then plated to a microplate with starting optical density 0.1 and treated with antiviral from 0.1–100 μg/mL (Supplementary Data 5) to ensure resistance, observed by uninhibited growth of the bacteria in antiviral-treated wells.

Multidrug-resistant *E. coli* used for positive control experiments (ATCC BAA-2471) were grown in Difco Nutrient Broth (BD catalog number 234000) with the addition of 25 μg/ml imipenem (MedChemExpress catalog number HY-B1369/CS-W019618) per manufacturer recommendations to prevent loss of the New Delhi metallo-beta-lactamase (NDM) plasmid.

**Measuring bacterial growth and inhibition**. Bacterial growth—measured as optical density (OD)— was monitored at different time points in the growth curves of *E. coli* and *B. cereus* to better understand the influence of antiviral drugs through the phases of bacterial growth. OD is a common approach to monitoring bacterial growth in culture[89]. We assessed the correlation between OD and colony forming units (CFUs) on untreated bacteria as well as across conditions challenging bacteria with both antiviral drugs with antibacterial properties and antiviral drugs with no putative antibacterial properties, and we found OD significantly correlated to CFU (Supplementary Data 2). Since using OD allowed for measurements in high-throughput manner, we used OD (absorbance 600 nm) as an indicator of bacterial growth for monitoring the impact of antiviral drugs on bacteria in culture.

Optical density (OD) measured at 600 nm was used to measure bacterial growth for *E. coli*, *B. cereus*, and BAA-2471. To measure growth after culturing in T25 flasks, OD was measured using Amersham Biosciences Ultrospec 3100 *pro* UV/Vis Spectrophotometer set to read absorbance at 600 nm. The OD of bacteria treated with antivirals or antibiotics was measured over 24 h at temperature set to 25 °C using a Biotek Synergy H1 multi-mode microplate reader equipped with Gen5 version 3.09 software.

**Growth assays**. Microplate-based assays were used to investigate the effects of antivirals and antibiotics on bacteria. For all experiments, 5 mL of LB, TSB, or Difco nutrient broth in a T25 vented flask was inoculated with a loop of bacteria taken from frozen glycerol stocks of single colony isolates. Cultures were grown for 4–5 h in an incubator set to 37 °C. After initial incubation, the OD of cultures was measured, and bacteria were diluted to OD 0.1. Using multichannel pipets, 180 μL of bacteria diluted to OD 0.1 was added to each well of sterile 96-well microplates (BD Falcon 353072). For each well, 20 μL of antiviral or antibiotic and MilliQ water (vehicle control) was added to reach 200 μL volume in each well. Using the Biotek Synergy H1 microplate reader, a set protocol was developed and applied to read microplates for each experiment. Plates were read directly after adding all reagents (bacteria, antivirals or antibiotics) to measure timepoint zero (T0), first shaking the plate in the microplate reader for 1 min at 567 cpm. The plate reader was then set to automatically run for up to 18 h, reading the OD (600 nm) of each well every hour and otherwise shaking the plate constantly with linear shake 567 cpm. Prior to starting the experiment, the plate reader was adjusted to 25 °C and held at 25 °C for the duration of the experiment. Since 25 °C is a relevant temperature for aquatic systems such as wastewater treatment plants and surface waters[90,91], this temperature was chosen for the experimental conditions. After running the plate reader constantly for at least 16 h to capture OD measurements for the full log phase of bacterial growth, the microplate was removed and transferred to a benchtop incubator (New Brunswick Scientific Incubator Shaker I2400) set to 25 °C and shaking at 100 RPM until 24 h at which point another reading for OD was taken on the microplate reader. Directly prior to reading the plate at 24 h, the microplate was first set in the microplate reader for linear shake for 1 min at 567 cpm. Antibiotic concentrations for antibiotic cross-resistance assays were chosen based on published minimum inhibitory concentration (MIC) values and

from experimentally assessing the range of growth response for each resistant strain of *E. coli* and *B. cereus* in the presence of a range of antibiotic concentrations.

**Statistics and reproducibility**. Raw data of absorbance values from the microplate reader were exported as an Excel document and then pasted into Prism (Prism 9 for macOS). Statistical analyses were performed in Prism and Excel. To test the significance of antibacterial effects of antivirals on *E. coli* and *B. cereus*, individual *t*-tests were conducted for each time point and antiviral concentration based on raw absorbance values of triplicate treated or untreated wells of bacteria from 96-well microplates. Prism was used to run unpaired *t*-tests with Welch correction, assuming individual variance for each group (treated vs. untreated triplicate wells). Differences were determined significant if the *p*-value met the threshold ($p > 0.005$ considered not significant, $p < 0.005$, $p < 0.0005$, or $p < 0.00005$).

Antiviral cross-resistance of antiviral-resistant *E. coli* and *B. cereus* was determined by comparing the growth of antiviral-treated and untreated antiviral-resistant strains and wild type bacteria. All mutant strains and wild type were challenged with 50 μg/mL antiviral for 24 h, which represented the time through logarithmic growth phase into stationary phase. For calculations comparing the growth of untreated and drug-treated wild type and resistant strains, the absorbance values at the 16-h time point were used, representing the time of peak growth or the end of log phase. Percent growth of antiviral-treated bacteria compared to untreated bacteria was calculated based on averaging the raw absorbance values across triplicate conditions for all strains and wild type tested. Antivirals in DMSO were compared to DMSO vehicle control in the "untreated" condition, and antivirals in water were compared to water vehicle control in the "untreated" condition. Percent growth was calculated using Excel.

Similarly, antibiotic cross-resistance of antiviral-resistant *E. coli* and *B. cereus* was determined by comparing the growth of antiviral-treated and untreated antiviral-resistant strains and wild type bacteria. Mutant strains and wild type were challenged with antibiotics at pre-determined concentrations based on MIC and variation in response across wild type and resistant strains. The percent growth of antibiotic-treated bacteria compared to untreated bacteria was calculated based on averaging the raw absorbance values at the 16-h time point across triplicate conditions for all strains and wild type tested. Minimum and maximum ranges for antibiotic effects were calculated by comparing the minimum and maximum absorbance values from triplicate conditions of antibiotic-treated and untreated wells. Percent growth relative to untreated control was calculated using Excel (Supplementary Data 10).

**Whole genome sequencing**. DNA was extracted from wild type and antiviral-resistant *E. coli* grown to log phase in 37 °C in LB or TSB media. Bacterial genomic DNA was isolated from cell pellets using the PacBio/Circulomics Nanobind CBB kit (102-301-900) and following the manufacturer's protocol for gram-negative bacteria. DNA samples were sheared with Covaris g-Tubes (Covaris, Woburn, MA) to an average size of 10–15 kb. Barcoded libraries were prepared using SMRTbell Express Template Prep Kit 2.0 (Pacific Biosciences, Menlo Park, CA). After adapter ligation, libraries were pooled in equimolar amounts and then size-selected using a BluePippin instrument (Sage Science, Beverly, MA) with a 10 kb cutoff. The library pool was bound to polymerase with the Sequel II Binding Kit 2.2. Sequencing was performed with the Sequel II Sequencing Kit 2.0 and an 8 M SMRT Cell on a Sequel II instrument (Pacific Biosciences, Menlo Park, CA). Resulting data were demultiplexed using the Lima utility (SMRT Link 10.2) and de novo genome assemblies generated using Canu v2.2 and the PacBio microbial assembly pipeline (SMRT Link 10.2). Both *B. cereus* and *E. coli* were sequenced; however, a contaminated *B. cereus* sample prevented us from being able to make genomic comparisons among *B. cereus* strains, and it was therefore left out of the analysis. Total base pair lengths of single assembled contigs are reported in Supplementary Data 12. Annotation was performed using the GALES prokaryotic annotation pipeline (freely available: https://github.com/jorvis/GALES). Genomes were deposited to GenBank under BioProject identification PRJA907821 and include accession numbers: CP115180 (stavudine-resistant *E. coli B*), CP115179 (lamivudine-resistant *E. coli B*), CP11578 (wild type reference *E. coli B*), CP117469 (zidovudine-resistant *E. coli B*), CP117043 (raltegravir-resistant *E. coli B*), CP117044 (dolutegravir-resistant *E. coli B*), CP115968 (didanosine-resistant *E. coli B*).

**Whole genome comparative analysis**. Assembly statistics confirmed a single contig for each genome sequenced and the total length of each genome as expected for *E. coli* (~4.6 million base pairs) (Supplementary Data 12). Comparative analysis of the sequenced genomes revealed unique single base pair insertions, deletions and substitutions, as well as multi-base pair insertions and deletions relative to the wild-type strains (Fig. 6, Fig. 7, Supplementary Data 13). Some mutations were found to be conserved across all sequenced antiviral-resistant mutants, which we concluded were collectively a unique mutation event in the wild-type clone during selection for sequencing (Supplementary Data 13). All mutations in each antiviral-resistant *E. coli* genome were identified based on comparative analysis with *E. coli B* wild type isolate sequenced (accession number CP11578) in the same run as the antiviral-resistant strains. Custom Perl scripts and BLAST (version 2.13.0+) were used to adjust the starting position and orientation of each annotated antiviral-

resistant genome, as the starting positions and orientations for each genome were not identical in the raw data. The gene coordinates were then numbered based on the unified starting position for all genomes. Codon numbers within each gene (identified using SnapGene Viewer) were used to describe the location of the identified mutations listed in Figs. 6,7. Mummer (4.0beta2) and DNASTAR MegAlign Pro (Version: 17.4.2) was used to identify genetic differences between wild type and antiviral-resistant genomes. Any base pair differences among the antiviral-resistant genomes compared to wild type were highlighted by the analysis. DNASTAR MegAlign Pro and SnapGene Viewer (version 6.1.1) were then used to cross-check each identified base pair mutation, comparing the sequence of wild type and antiviral-resistant genomes. Translations of amino acid sequences were observed in SnapGene Viewer, and protein identity and sequence for the gene containing the mutation were confirmed again using blastx (https://blast.ncbi.nlm.nih.gov/Blast.cgi) entering the FASTA sequence and restricting the search to *E. coli* for organism (taxid:562). Truncation events were identified in antiviral-resistant genomes when, in comparison to wild type *E. coli B* as well as nucleotide sequences from other *E. coli* (taxid:562) deposited to GenBank and retrieved using blastx, the antiviral-resistant gene was shortened after an early stop codon following an identified mutation. Percent protein truncated due to mutation, reported in Figs. 6,7, was calculated based on comparison to wild type *E. coli B* gene length after the stop codon in the antiviral-resistant genome. For all mutations reported in Figs. 6, 7, stop codons were observed after the mutation due to a frame shift in the gene introduced by the mutation. Stop codons were identified by observing the three sequential nucleic acids TAG, TAA, or TGA. The on-line Resistance Gene Identifier (RGI 6.0.1; https://card.mcmaster.ca/analyze/rgi) was used to identify genes within reference and mutant genomes matching the Comprehensive Antibiotic Resistance Database (CARD 3.2.6) restricting to perfect and strict matching criteria. Differences in gene sequences within antibiotic resistance genes were identified using a custom perl script. We also searched the literature for evidence of a role in antibiotic resistance for genes that contained the most significant changes to the protein coding genes. All scripts used to compare the genomes of *E. coli B* wild type and antiviral-resistant *E. coli* strains are freely available (https://github.com/spacocha/mutant_strain_comparative_genomics).

**Reporting summary**. Further information on research design is available in the Nature Portfolio Reporting Summary linked to this article.

## Data availability

Source data for Figs. 3, 4, and 5 can be found in Supplementary Data. All genomes included in the study were deposited to GenBank under BioProject identification PRJA907821 and include the following accession numbers: CP115180 (stavudine-resistant *E. coli B*), CP115179 (lamivudine-resistant *E. coli B*), CP11578 (wild type reference *E. coli B*), CP117469 (zidovudine-resistant *E. coli B*), CP117043 (raltegravir-resistant *E. coli B*), CP117044 (dolutegravir-resistant *E. coli B*), CP115968 (didanosine-resistant *E. coli B*).

## Code availability

All scripts used to compare the genomes of *E. coli B* wild type and antiviral-resistant *E. coli* strains are publicly available on the following site: https://github.com/spacocha/mutant_strain_comparative_genomics.

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

## Acknowledgements
Whole genome sequencing and annotation were completed by Maryland Genomics at the Institute for Genome Sciences, University of Maryland School of Medicine, and we especially thank Luke Tallon and Lisa Sadzewicz for their advising on the use of the PacBio Sequel II platform. We acknowledge funding from the Johns Hopkins University Fisher Center Discovery Program. We thank Marsha Wills-Karp and Anthony So for their feedback on the original manuscript.

## Author contributions
C.P. and E.G.S. conceived of the initial study and contributed to study design, data interpretation and analysis. V.J.W. contributed to study conception and design, performed all experiments and data collection, performed data analysis, and drafted the manuscript. E.G.S. and S.P.P. contributed to manuscript drafting. S.P.P. contributed to data interpretation and study design, wrote custom scripts, completed whole genome alignment and base pair mutation identification, and conducted CARD search and analysis. C.P. provided funding. All authors were involved in revising the manuscript.

## Competing interests
The authors declare no competing interests.
