## [Peer Review File · Communications Biology]

Reviewers' comments:

Reviewer #1 (Remarks to the Author):

The present study by Wallace et al. is focused on the ever-growing health threat of antimicrobial resistance. In this study, certain antivirals have been shown to inhibit bacterial growth, thereby acting as antibacterial agents. Further, these antivirals modulate bacterial resistance and at times elicit cross-resistance against antivirals from different classes and, interestingly, even antibiotics and phages.

While this is an interesting, well-written study, albeit in need of major adjustments, it does not fit into the scope of the journal and would be better suited in a specialized journal more geared toward antibiotic resistance.

Major revisions:

1. Lines 106-108: Has there been any evidence or studies on this topic or is this the authors' hypothesis?
2. Lines 113-115: A study showed that ZDV prevents the transfer of antibiotic resistant genes, (Buckner et al., HIV Drugs Inhibit Transfer of Plasmids Carrying Extended-Spectrum β -Lactamase and Carbapenemase Genes. *mBio*. 2020). How would the authors argue this finding, as it is in contradiction with the present study?
3. Lines 128-129: How were the concentrations of the drugs decided? Is the study implicated for gut concentrations or blood concentrations?
4. Lines 132-135: Why did the authors repeat the experiments which have been already shown in the papers by Shilaih et al, 2018 and Ray et al, 2021? Is there any rationale behind this experiment?
5. Lines 141-143: Which experiment was done to show the bacterial inhibition against antiretrovirals? The authors haven't mentioned it in these lines. Fig S2 shows the changes in absorbance of bacteria when treated with ZDV, however dead bacteria also give absorbance values. I would suggest doing cell viability experiment (CFU), to understand the sensitivity of antiviral exposure. Also, in line 48 it is mentioned a 48-hour timeframe, which should result in a great amount of dead bacteria. CFU readings will show a better viability comparison.
6. Fig. S2 Why did the authors used 25°C for incubation? Is the resistance plasmid-induced? Do the authors have any explanation for this temperature-dependence?
7. Lines 166-167: Please mention the growth rates, so that it's clear to the readers that there is no change in the growth curves between the resistant and susceptible strains.
8. Line 168: Can the lower growth of ZDV-resistant bacteria be due to bacteriostatic concentration of ZDV used? Usually, bacteria can be grown at sub-MIC concentration (lower than the MIC), and then the bacteria can be plated at >MIC so that only the colonies with better growth can be selected. Further, the colony can be grown with/without ZDV to check if the resistance is stable. Can the authors do a growth curve after and share the resulting growth kinetics?
9. Line 209: Can the authors try the same method for maintaining stable resistance for DTG-resistant strain of *B. subtilis*?
10. The wild-type strain shows resistance in some cases, like in gentamicin for *E. coli*. Have the authors tried using any other isolate of *E. coli* or *B. subtilis*?
11. The authors haven't used any positive control like an MDR *E. coli* in Fig 4.
12. Was any phage resistance observed in *B. subtilis*?
13. ZDV is stable at 37 °C and so are the bacteria. However, the survival of ZDV-resistant strain is compromised at that temperature. Therefore, it seems that the mutation is not stable. Have the authors checked this resistance in different generations?

General revisions

14. A study has shown that ZDV can act against colistin-resistant *E. coli* (Peyclit L et al., Zidovudine: A salvage therapy for *mcr-1* plasmid-mediated colistin-resistant bacterial infections? *Int J Antimicrob Agents*. 2018). Does this suggest that ZDV-resistant bacteria can also be resistant to beta lactam

antibiotics? Can the authors throw some light on this?

15. The discussion has a lot of repetition of results.

16. In line 368, the authors mention about mutations causing for antiviral resistance. However, it is also known that certain antivirals like TDF have been metabolized by *G. vaginalis* (Klatt et al., Vaginal bacteria modify HIV tenofovir microbicide efficacy in African women. *Science*. 2017). What would the authors comment?

17. Antiretrovirals like ZDV are known to affect the DNA replication in bacteria, similar to trimethoprim. Could it be a possibility that ZDV-resistant bacteria was resistant to trimethoprim due to similar targets?

18. Mutations in the bacteria due to antiviral might be the reason for antibiotic susceptibility or resistance. What is the clinical implication of this study?

19. In diseases such COVID-19 or HIV infection, antibiotics was administered to prevent any associated bacterial infections. How would the authors explain viral resistance with exposure to such antibiotics and should the treatment of individuals be supplemented with other strategies?

20. Additionally, in a host, in the presence of both antivirals and antibiotics, how would the resistance development be affected in a bacterium?

Reviewer #2 (Remarks to the Author):

In this study Wallace et al tackle the question of the effect of antiviral drugs on bacterial growth and resistance to antibiotics and phages. They took 14 antiviral drugs with different modes of actions and exposed two different bacteria (*E. coli* and *B. subtilis*) to ranging concentrations these drugs. After observing growth effect on some of these, they isolated resistant ones and then analyzed patterns of cross resistance of these resistant clones to the anti-viral drugs, antibiotics and phage T4. They conclude on the ability of anti-viral drugs to induce antimicrobial resistance.

I found the initial idea of the study proposed by Wallace and colleagues original and timely. Indeed, both a renewed interest in anti-viral drugs and cross resistance mechanisms makes the subject of this study original. However, I felt the authors could go quite deeper to go beyond a very descriptive study to be able to really understand how their findings fit in the the current context on antibacterial resistance.

Major Points

- To be able to start understanding if how the authors suggest "a unique mechanism of resistance may emerge" (1388), a necessary experiment for me is to sequence a few clones to put mutations they observe into context. Are they different from commonly observed mutations in antibacterial or phage resistance? What is expected in that regard

- Very little information is given about the strains chosen for the experiments. Which strains of *E. coli* and *B. subtilis*? Why these ones? Are the authors confident that such results would be extendable to other strains from the same species?

- The phage aspect of the paper is very weak in terms of experiments. It is well known that growth has a huge impact of phage host interaction. Thus the observed effects could be due to change in growth. More over it is not precised at which multiplicity of infection the experiments are performed (an absolute number of phage is given, but not MOI which is the relevant parameter), nor when the phages were added. Usually, phages are added during exponential growth at OD=0.3. Overall, I would suggest that the authors 1) Perform their experiments at different MOI by adding the phages in exponential phage growth. This would allow to also assess how strong the resistance is (might be for low MOI but not for high). 2) The authors confirm these results by doing similar experiments in solid media (plaque assays experiments).

- The cross resistance claim regarding antibiotics could also be more substantiated. While some effect was observable, it was sometimes more susceptible, sometimes more resistance, without any consistent observations in terms of mechanisms or modes of actions of antibiotics or antivirals. Could these be more "random" side effects? Also, maybe I missed it, but I did not see if all concentrations

were tested and if yes, I did not find the results in the supp material, and how were the ones put forward chosen for main figure. If not, how were these chosen ?

- In the discussion, I feel the limits of the study were not properly discussed. The discussion is already very long, I would suggest to shorten it, especially as many of it is very speculative. While it's quite interesting to speculate and there can be space dedicated to it, first limits of the study should be discussed. I was typically wondering, when adding other types of bioactive compound (not anti-viral but of other types) how many would be expected to cause such effects that we observe ? So are the authors in the study underlying a potential specific mechanism from anti-viral drugs, or just finding side effects of adding bioactive compound to cellular cultures ?

- Throughout the study, I would have enjoyed more thoughts/hypothesis on potential mechanisms that could explain the observed patterns of cross resistance (like the fact that most mechanisms led to broad resistance to other anti-viral compounds) and also given that mechanisms of anti viral actions and antibiotics are known what could be expected at the "crossroad".

Minor points

- While I am not familiar with anti-viral drugs, which might be the case of many readers of the study, it would be of help, in the introduction or in the text to give an order of magnitude at which each of these drugs are used, to estimate whether the tested concentrations are within interesting biological range.

- I found the growth curves presented in S2 quite more informative than the summary of Figure 1. Maybe to keep the summary of figure 1 but adding to that figure, key growth curves. For example the one with zidobuvine was very convincing while some of the differences for others were much more subtle / less conclusive

- Figure S2: experiments were done in triplicates, but what is represented there ? One out of the three experiments ? Means ? In any case, it would be good to get a sense of the variation (maybe each curve could have a shadow representing the variations between the replicates). Also n=3, means biological replicates ?

- Figure S5: It is not explained what is "color key". Not clear to me what red or green is. Also not explained how the hierarchical clustering was done.

- Figure 3: did not really understand how the statistics were performed

- There was no specific method section dedicated to phage, so could not assess it. How was it amplified ? titrated ? When were the phages added to growing bacteria ?...

16 December 2022

We thank the reviewers for the valuable comments on our manuscript and for allowing us to resubmit. We were able to significantly improve the manuscript based on the reviewer comments.

In the revision process of our manuscript, we have made several additional major revisions to the work. Please kindly note the following:

- We have excluded all experiments involving bacteriophage. In place of the bacteriophage work, we have included results from whole genome sequencing of antiviral-resistant *E. coli* isolates. We have integrated the sequencing results and discussion into our previous analyses on the phenotypic characterization of antiviral-resistant *E. coli* and *B. cereus*.
- During sequencing, we discovered the *B. subtilis* we had received was misidentified in our original manuscript. We have corrected the work and properly described our results with *B. cereus*. Our goal to provide evidence of antiviral resistance in a model gram-positive microorganism has still been achieved.
- We report on a zidovudine-resistant *E. coli* strain that does not exhibit temperature-dependent growth as observed previously. The temperature-dependent growth of zidovudine-resistant *E. coli* was a cause of concern for the reviewers in our original submission; we therefore went forward ensuring that temperature-independent zidovudine-resistant *E. coli* could be developed and isolated by the same methods employed for the development of all other antiviral-resistant strains. We have included this note in the relevant reviewer comments and replaced all relevant data, results and discussion throughout the manuscript with data on the new zidovudine-resistant strain.

Since we removed our results and discussion on bacteriophage, point (3) listed above is not addressed in our revised manuscript. We have also therefore not addressed comments from Reviewer 1 and Reviewer 2 referencing the bacteriophage work. Aspects on the mechanisms of resistance for drugs but not phage (point (1) above) are addressed. Cell viability (2), stability of mutations (4), the possibility of “random” side effects (5), and all comments (6) are discussed in detail below. Please do let us know if we can provide any additional materials or details at this time. We appreciate your consideration of our manuscript and its revisions.

We have *italicized and underlined reviewer comments* on the original manuscript.

REVIEWER #1

Major revisions:

1. Lines 106-108: Has there been any evidence or studies on this topic or is this the authors' hypothesis?

This is the authors' hypothesis. The evidence we provide in the manuscript is that bacteria can become resistant to antibacterial antiviral drugs and develop cross-resistance to antibiotics. The statement is a hypothesis based on our findings. The original lines 106-108 have been rewritten, and we have included language at the end of the paragraph to emphasize that additional research is needed on this subject:

Lines 89-94: However, an understanding of the full spectrum of contributors to antimicrobial resistance remains incomplete. Non-antibiotic pharmaceuticals such as antivirals may also contribute to the development of antimicrobial resistance on a global scale, but research is needed to gain full understanding of how drugs with antibacterial properties can lead to mutations in bacteria and cause cross-resistance to antibiotics.

2. Lines 113-115: A study showed that ZDV prevents the transfer of antibiotic resistant genes, (Buckner et al., HIV Drugs Inhibit Transfer of Plasmids Carrying Extended-Spectrum β -Lactamase and Carbapenemase Genes. mBio. 2020). How would the authors argue this finding, as it is in contradiction with the present study?

Thank you for mentioning the Buckner et al. paper. It is true the transfer of resistance genes is one means by which bacteria can acquire resistance. In our study, we assessed the development of resistance due to single antiviral drug exposures in monoculture. We observed that *E. coli* exposed to various antiviral drugs including zidovudine, studied in the Buckner et al. paper, can develop resistance to the exposure drug. As we demonstrated, the resistance that develops can confer cross-resistance to antibiotics and other antiviral drugs. Bacteria can develop several resistance mechanisms in response to an antibacterial agent. It is quite possible that zidovudine could be capable of reducing plasmid transmission, as described in Buckner et al., while simultaneously leading to the development of resistance mechanisms such as upregulation of efflux pumps, alterations in metabolic pathways, or target modification in bacteria.

To clarify that we are only considering mutations that arise *de novo*, rather than arising due to plasmid transfer, we have referenced the Buckner et al. paper and included the following line:

Line 558-562: In other contexts, zidovudine has been shown to prevent the transfer of antibiotic resistance genes by horizontal gene transfer⁸². Our studies provide evidence that chromosomal mutations that confer cross-resistance to the antiviral zidovudine and naïve cross-resistance to antibiotics can develop *de novo* due to the presence of the exposure drug.

3. Lines 128-129: How were the concentrations of the drugs decided? Is the study implicated for gut concentrations or blood concentrations?

The concentrations of the drugs in the lines referenced were decided upon based on published literature-- studies that have established circulating concentrations of the drugs used in our experiments, studies that have investigated inhibitory concentrations of some of these drugs on bacteria, and studies that have determined concentrations of the drugs in the aquatic environment. The study would be implicated for both gut and blood concentrations, as well as concentrations in the aquatic environment. We have improved the text by adding the following lines to address how the concentrations of drugs were decided and the relevance:

Lines 132-133: The concentration range tested covers inhibitory and sub-inhibitory concentrations demonstrated in previous *in vitro* tests with bacteria¹⁰⁻¹².

Line 135-137: The concentration range tested also includes therapeutic or circulating concentrations of antiviral drugs, such as the plasma concentration of zidovudine (0.016-1.7 mg/L)³² and therapeutic window of efavirenz (1-4 mg/L)³³.

Lines 575-576: Many antivirals persist through wastewater treatment⁹⁰⁻⁹³ and are released into the environment, detectable at concentrations ranging from ng/L to µg/L.

4. Lines 132-135: Why did the authors repeat the experiments which have been already shown in the papers by Shilaih et al, 2018 and Ray et al, 2021? Is there any rationale behind this experiment?

We repeated the studies to serve as a control, to demonstrate that we could reproduce the results observed in Shilaih et al. and Ray et al. Reproducing what has been shown previously allowed us to validate our methods and then extend the investigation of antiviral drugs' impact on bacterial growth. For example, we investigated how different concentrations of drug affect growth curves at different points in growth phases, which may give an indication about their mechanism of action. No previous studies, to our knowledge, have investigated the antibacterial effects of dolutegravir. We have incorporated the explanation of our rationale with the following lines:

Lines 133-135: We included concentrations tested in existing literature to ensure reproducible results and to validate our methods.

Lines 138-141: However, in attempt to establish inhibitory values for various incubation periods and exposure concentrations, the concentration ranges tested generally exceeded environmentally detected levels or a typical plasma concentration in the human body³⁴.

5. Lines 141-143: Which experiment was done to show the bacterial inhibition against antiretrovirals? The authors haven't mentioned it in these lines. Fig S2 shows the changes in

absorbance of bacteria when treated with ZDV, however dead bacteria also give absorbance values. I would suggest doing cell viability experiment (CFU), to understand the sensitivity of antiviral exposure. Also, in line 48 it is mentioned a 48-hour timeframe, which should result in a great amount of dead bacteria. CFU readings will show a better viability comparison.

We thank the reviewer for these questions and suggestions. We have changed the text to clarify that Figure 1 shows the results of bacterial inhibition in the presence of antiviral drugs:

Lines 161-164: Growth inhibition, as seen by a significant reduction in OD, was observed within four hours of exposure for several of the antivirals with antibacterial activity against *E. coli*, and all antivirals with antibacterial activity inhibited growth within eight hours (Fig .1).

As suggested, we completed the CFU experiments. Table S2 contains detailed experimental data, and lines 141-150 describe the results. We further considered the 48-hour time point was unnecessary beyond 24 hours, and therefore we only included data for all experiments up to 24 hours.

Lines 141-150: Bacterial growth –measured as optical density (OD)-- was monitored at different time points in the growth curves of *E. coli* and *B. cereus* to better understand the influence of antiviral drugs through the phases of bacterial growth. OD is a common approach to monitoring bacterial growth in culture³⁵. We assessed the correlation between OD and colony forming units (CFUs) on untreated bacteria as well as across conditions challenging bacteria with both antiviral drugs with antibacterial properties and antiviral drugs with no putative antibacterial properties, and we found OD significantly correlated to CFU (Table S2). Since using OD allowed for measurements in high-throughput manner, we used OD (absorbance 600 nm) as an indicator of bacterial growth for monitoring the impact of antiviral drugs on bacteria in culture.

6. Fig. S2 Why did the authors used 25°C for incubation? Is the resistance plasmid-induced? Do the authors have any explanation for this temperature-dependence?

Thank you for recommending we highlight our rationale for 25°C. Twenty-five degrees Celsius is an environmentally-relevant temperature for surface water and wastewater influent and effluent, and this is ultimately the context we are interested in. We have included the information in lines 661-664 to highlight this for the reader:

Lines 661-664: Prior to starting the experiment, the plate reader was adjusted to 25°C and held at 25°C for the duration of the experiment. Since 25°C is a relevant temperature for aquatic systems such as wastewater treatment plants and surface waters^{96,97}, this temperature was chosen for the experimental conditions.

Caption Figure S2: Growth of (a) *E. coli* and (b) *B. cereus* in the presence of antivirals, measured as absorbance (600nm) over 24 hours in 25°C incubation, as relevant temperature for surface waters and wastewater conditions.

In the revised version of this manuscript, we have included data for a zidovudine-resistant strain that is not temperature-dependent. Since the mutations we observed are not likely to be acquired by horizontal gene transfer, the resistance is not likely to be plasmid-induced.

7. Lines 166-167: Please mention the growth rates, so that it's clear to the readers that there is no change in the growth curves between the resistant and susceptible strains.

Thank you for letting us know the reader may benefit from knowing the growth rates. We have included these in Table S4. We have mentioned the growth rates in the following lines:

Lines 190-198: Resistant strains achieved similar maximum growth and growth kinetics compared to wild type, with the exception of the *E. coli* zidovudine-resistant strain which achieved 32% lower maximal growth compared to wild type (Table S4; Fig. S3). During the incubation period of 4-12 hours, while bacteria were in log phase and OD was rapidly increasing, wild type *B. cereus* showed an average increase in OD of 0.162 (absorbance, 600 nm) per hour, and the antiviral-resistant strains showed an increase of 0.154 per hour. Wild type *E. coli* showed an OD increase of 0.162 per hour, and antiviral-resistant strains (excluding zidovudine-resistant *E. coli*) showed an average increase in OD of 0.199 per hour (Table S4). Resistance to antivirals was observed at sub-inhibitory concentrations, typically at < 100 µg/mL (Table S5; Fig. S4). Resistance was not observed in efavirenz treatments for both *E. coli* and *B. cereus*; resistance was observed in *E. coli* acyclovir treatments, but a resistant strain was not isolated (Table S5).

8. Line 168: Can the lower growth of ZDV-resistant bacteria be due to bacteriostatic concentration of ZDV used? Usually, bacteria can be grown at sub-MIC concentration (lower than the MIC), and then the bacteria can be plated at >MIC so that only the colonies with better growth can be selected. Further, the colony can be grown with/without ZDV to check if the resistance is stable. Can the authors do a growth curve after and share the resulting growth kinetics?

We understand the reviewer to be asking about whether zidovudine-resistant *E. coli* can be grown in >MIC concentration of zidovudine and whether the resistance is stable. All antiviral-resistant strains were isolated after exposure to drug and then grown in the absence of drug. This also ensured that the resistance was stable. All experiments to challenge antiviral-resistant strains and wild type with antiviral drugs and antibiotics were conducted in the absence of the initial exposure drug. That is, zidovudine-resistant *E. coli* were grown in the presence of zidovudine, passaged without zidovudine, later challenged with zidovudine and other antiviral drugs as well as antibiotic drugs and demonstrated resistance over 24 hours by exhibiting the same maximal growth and growth kinetics as the untreated condition. The growth kinetics for the zidovudine-resistant strain are shown in Figure S3. Further detail on the growth of zidovudine-resistant *E. coli* in the presence of >MIC concentration of zidovudine is included in Table S6. Table S7 includes the data on zidovudine-resistant *E. coli* passaged in the absence of drug 15 times and demonstrating stable resistance to zidovudine and antibiotics.

To clarify this within the text, we added the following lines:

Figure S3: Growth of untreated antiviral-resistant mutant strains compared to growth of untreated wild type *E. coli* and *B. cereus*, in the absence of drug treatment.

Lines 228-232: Antiviral-resistant strains were, overall, shown to have stable resistance phenotypes, as demonstrated over 15 passages. For example, dolutegravir-resistant *B. cereus* maintained resistance to raltegravir, dolutegravir, and efavirenz throughout passaging in the absence of dolutegravir; zidovudine-resistant *E. coli* maintained resistance to zidovudine throughout 15 passages in absence of zidovudine (Table S7).

9. Line 209: Can the authors try the same method for maintaining stable resistance for DTG-resistant strain of *B. subtilis*?

Thank you for suggesting we also try the same method for demonstrating the stable resistance of other strains. We have included these data in Table S7 and added lines 228-232 to clarify within the text:

Lines 228-232: Antiviral-resistant strains were, overall, shown to have stable resistance phenotypes, as demonstrated over 15 passages. For example, dolutegravir-resistant *B. cereus* maintained resistance to raltegravir, dolutegravir, and efavirenz throughout passaging in the absence of dolutegravir; zidovudine-resistant *E. coli* maintained resistance to zidovudine throughout 15 passages in absence of zidovudine (Table S7).

10. The wild-type strain shows resistance in some cases, like in gentamicin for *E. coli*. Have the authors tried using any other isolate of *E. coli* or *B. subtilis*?

We thank the reviewer for this point and agree it is an excellent suggestion for future studies. While we have not used other strains, we would like to highlight that our findings are a proof-of-principle for what is possible using model gram-negative and gram-positive bacteria. To recognize the importance of testing other strains of *E. coli* and *B. cereus* and to note our interest in expanding the work beyond these two model organisms, we have modified the following text:

Lines 589-595: In the future, isolating and challenging a large number (>5) of antiviral-resistant isolates from each antiviral exposure will help answer whether antiviral-resistance mutations are consistent or stochastic. Given the diversity of gram-positive and gram-negative bacteria, using additional strains of *E. coli* and expanding this work beyond *E. coli* and *B. cereus* will help elucidate the broader implications for the effects of antibacterial antivirals on bacterial communities such as those found in environmental waters, wastewater treatment plants, and the human gut.

11. The authors haven't used any positive control like an MDR *E. coli* in Fig 4.

Thank you for your suggestion to use a positive control such as multidrug-resistant *E. coli*. We have created an additional section of the manuscript using multidrug-resistant *E. coli* and included the results in Figure 4 and Table S11. The new text includes the following lines:

Lines 326-365: Following our results demonstrating that antiviral-resistant *E. coli* can exhibit broad cross-resistance to antibiotic drugs, we wanted to contextualize the clinical relevance of the survival and growth of antiviral-resistant *E. coli*. To validate the antibiotic cross-resistance phenotypes of the antiviral-resistant strains and the *in vitro* method of evaluation for resistance, we challenged a well-characterized multidrug-resistant *E. coli* strain (ATCC BAA-2471) with the same concentrations of antibiotics alongside the antiviral-resistant *E. coli* strains. *E. coli* BAA-2471 exhibits resistance to many classes of antibiotics including penicillins, cephalosporins, carbapenems, quinolones, aminoglycosides, and others³⁷. The strain is therefore resistant to antibiotics we tested including amoxicillin, gentamicin, tetracycline, trimethoprim and sulfamethoxazole. We confirmed the resistance of BAA-2471 to these antibiotics using our methods, and we also compared the strain's resistance or susceptibility to antiviral drugs (Fig. 4, Table S11). Zidovudine-resistant *E. coli*—the antiviral-resistant *E. coli* strain with the broadest resistance profile—and untreated BAA-2471 exhibited comparable growth kinetics. Resistance was indeed observed for all antibiotics to which BAA-2471 was purportedly resistant, and the growth of BAA-2471 and zidovudine-resistant *E. coli* in the presence of antibiotic treatment was comparable. While both BAA-2471 and zidovudine-resistant *E. coli* demonstrated growth in the presence of 3 µg/ml gentamicin, we found growth of BAA-2471 inhibited by 5 µg/ml gentamicin in our assays whereas zidovudine-resistant *E. coli* were not susceptible (Fig. 3, Table S11). In the presence of 50 µg/ml zidovudine, after 16 hours' incubation, zidovudine-resistant *E. coli* exhibited 120% growth compared to untreated control whereas BAA-2471 exhibited only 48% growth. These results both validated our approach to assessing the phenotypic resistance profiles of resistant strains and allowed comparative analysis of bacterial growth in the presence and absence of antiviral drugs and antibiotics using a well-characterized resistant clinical isolate. Further, the results suggest that zidovudine-resistant *E. coli* exhibit resistance mutations that are unique compared to the clinical isolate BAA-2471 yet grant comparable fitness in the absence of drug and comparable survival in the presence of antibiotic concentrations that are inhibitory for wild type.

12. Was any phage resistance observed in *B. subtilis*?

We have removed the results and discussion of phage upon revision.

13. ZDV is stable at 37 °C and so are the bacteria. However, the survival of ZDV-resistant strain is compromised at that temperature. Therefore, it seems that the mutation is not stable. Have the

authors checked this resistance in different generations?

We thank the reviewer for the concern on the stability of the mutation. We have checked that the resistance is in different generations and demonstrated the results in Table S7, which contains the raw data for demonstrating the stability of the phenotype across 15 generations. A new zidovudine-resistant strain was isolated and reported on for the revised version of this manuscript that was grown (stably) in 37 °C. All experimental data in this updated manuscript contain data using this strain. The following text clarifies that the strain and resistance mutation is stable across generations:

Lines 231-232: zidovudine-resistant *E. coli* maintained resistance to zidovudine throughout 15 passages in absence of zidovudine (Table S7).

General revisions

14. A study has shown that ZDV can act against colistin-resistant *E. coli* (Peyclit L et al., Zidovudine: A salvage therapy for mcr-1 plasmid-mediated colistin-resistant bacterial infections? *Int J Antimicrob Agents*. 2018). Does this suggest that ZDV-resistant bacteria can also be resistant to beta lactam antibiotics? Can the authors throw some light on this?

Thank you for mentioning the Peyclit et al. study. As described in our results throughout the manuscript, zidovudine-resistant *E. coli* showed resistance to amoxicillin, which is a beta-lactam antibiotic. As the question is posed, “does this suggest ZDV-resistant bacteria can also be resistant to beta lactam antibiotics” without specifying certain bacteria, we would like to note that beta-lactam antibiotics are broad spectrum antibiotics, and the gram-positive bacteria appear to be intrinsically resistant to nucleoside analogs including zidovudine. Therefore, the mechanism of resistance to zidovudine as well as other drugs is likely to vary among species and even strains of bacteria, and the implications of resistance to zidovudine and cross-resistance to a beta-lactam antibiotic is not necessarily linked in every case. Within a strain, resistance mechanisms could also be heterogeneous, and we have mentioned this in lines 595-597 as a course of future study. Zidovudine may also have multiple cellular targets for bacteria, and not every resistance mutation due to zidovudine exposure may result in beta lactam cross-resistance in gram-negative bacteria.

Lines 595-597: Mechanisms of resistance to antiviral drugs are likely to vary across species and even strains of bacteria, and the patterns of cross-resistance that arise may therefore also be heterogeneous.

15. The discussion has a lot of repetition of results.

Thank you for letting us know that the discussion has a lot of repetition of results. In order to improve the discussion, we have condensed our previous discussion on the phenotypic results of antiviral-resistant strains, removed discussion of phage-resistant *E. coli*, and included new discussion on whole genome sequencing results of *E. coli* isolates.

16. In line 368, the authors mention about mutations causing for antiviral resistance. However, it is also known that certain antivirals like TDF have been metabolized by G. vaginalis (Klatt et al., Vaginal bacteria modify HIV tenofovir microbicide efficacy in African women. Science. 2017). What would the authors comment?

Thank you for mentioning the Klatt et al. study. Metabolism is one means by which bacteria can handle a xenobiotic. Indeed, it is well-established that bacteria metabolize various pharmaceutical compounds, particularly in wastewater treatment settings (e.g., Ahmed et al., *Progress in the biological and chemical treatment technologies for emerging contaminant removal from wastewater: A critical review. 2017*). In reference to our manuscript, we are discussing how antiviral drugs with antibacterial properties exert pressure on bacteria such that they mutate and develop resistance mutations in response to the exposure. For the study mentioned, TDF is not exerting observable antibacterial effects as the bacteria are observed to metabolize the drug.

17. Antiretrovirals like ZDV are known to affect the DNA replication in bacteria, similar to trimethoprim. Could it be a possibility that ZDV-resistant bacteria was resistant to trimethoprim due to similar targets?

Yes, thank you for this observation. We believe this could certainly be the case as well. We have provided the additional detail in the following lines:

Lines 557-558: Trimethoprim and sulfamethoxazole, specifically, target tetrahydrofolate metabolism, which also implicates thymidine and thymidine kinase⁸².

18. Mutations in the bacteria due to antiviral might be the reason for antibiotic susceptibility or resistance. What is the clinical implication of this study?

Further research is certainly needed to understand the full scope of clinical implications of antiviral use, but our research is the first to demonstrate the possibility. In the introduction, we emphasize the importance of studying the impacts of non-antibiotic drugs on antimicrobial resistance:

Lines 86-94: In the United States alone, there are over 2.8 million antibiotic-resistant infections every year and over 35,000 deaths due to antibiotic resistance¹⁷. Most discussion on antimicrobial resistance control centers the judicious use of prescription antibiotics and hygiene^{18,19}. However, an understanding of the full spectrum of contributors to antimicrobial resistance remains incomplete. Non-antibiotic pharmaceuticals such as antivirals may also contribute to the development of antimicrobial resistance on a global scale, but research is needed to gain full understanding of how drugs with antibacterial properties can lead to mutations in bacteria and cause cross-resistance to antibiotics.

We would like to highlight the final lines of the manuscript for the global clinical implications (Lines 602-604): “The risk of developing antimicrobial resistance due to the presence of antivirals may also be highest where the burden of viral disease –and concurrent use of antiviral drugs– is also highest.”

19. In diseases such COVID-19 or HIV infection, antibiotics was administered to prevent any associated bacterial infections. How would the authors explain viral resistance with exposure to such antibiotics and should the treatment of individuals be supplemented with other strategies?

This is an interesting thought and certainly one that warrants investigation. While viral resistance is beyond the scope of this manuscript, a future study could incorporate this as well as investigate the clinical feasibility of amending treatment protocols given the findings.

20. Additionally, in a host, in the presence of both antivirals and antibiotics, how would the resistance development be affected in a bacterium?

This is another interesting thought, and we thank the reviewer for such interest in our work. Depending on the mechanisms of action and pathways affected, antiviral drugs and antibiotic drugs may exert synergistic or antagonistic effects on bacteria. They may also act through different pathways. The myriad of possibilities given different combinations and mixtures – particularly complex mixtures such as environmental water samples—highlights both the difficulty in predicting the effects on a bacterium and the need to investigate such questions. While we were not able to explore the interactions of antibiotic-antiviral mixtures on the outcome of resistance within the scope of this manuscript, we are very interested to pursue these questions further.

REVIEWER #2

Major Points

1. To be able to start understanding if how the authors suggest “a unique mechanism of resistance may emerge” (1388), a necessary experiment for me is to sequence a few clones to put mutations they observe into context. Are they different from commonly observed mutations in antibacterial or phage resistance? What is expected in that regard

Thank you for your suggestion to sequence a few clones. We have removed the aspect of phage and phage resistance in our manuscript revisions. However, we discuss the results of sequences in section 2.4 *Comparative genome analysis of antiviral-resistant E. coli* (lines 366-468). While some mutations such as the multidrug efflux were observed and align with commonly observed mutations in antibacterial resistance, other mutations have not been previously reported in the literature as being associated with antibacterial resistance. We could hypothesize that this is expected, as some antiviral drugs may act on similar targets as antibiotics, but other antiviral drugs may have unique targets. The following lines added in the discussion also address this aspect:

Lines 515-530: Many of the observed genetic mutations in antiviral-resistant isolates occurred in proteins or pathways with no known role in antimicrobial resistance. However, many of the genes in which the mutations occurred are known to be involved in aspects of metabolism and nutrient transport such as galactofuranose-binding protein, sodium/glutamate transporter, and acetate coenzyme A-transferase. Consistent with the existing literature that suggest an array of metabolic pathways may be implicated in antimicrobial resistance or may provide targets for novel therapies for multidrug-resistant bacterial infections^{26,39,58,59,61}, these mutations may still point to key genes involved in resistance to antiviral drugs and associated antibiotic cross-resistance. Many genes beyond those traditionally thought to be responsible for antibiotic resistance have been highlighted as relevant for clinical isolates with antibiotic resistance, such as mutations in N-acetylglucosamine deacetylase⁷², glycerol kinase²⁷, glycosyltransferase⁷³, serine-threonine kinase⁷⁴, and histidine kinase⁷⁵. Mutations in the PTS and in *crp* (Table S13), such as those observed for dolutegravir-resistant *E. coli*, have been shown to be implicated in broad tolerance or resistance to antimicrobials^{26,76,77}. The mutations observed in antiviral-resistant genomes may therefore provide further support that mutations in genes beyond traditional resistance genes may be implicated in a range of antimicrobial resistance and cross-resistance to antibiotic drugs.

2. Very little information is given about the strains chosen for the experiments. Which strains of *E. coli* and *B. subtilis* ? Why these ones ? Are the authors confident that such results would be extendable to other strains from the same species ?

We have included the following line at the beginning of the results to specify which strains:

Line 129-131: Gram-negative *E. coli* and gram-positive *B. cereus* were chosen as well-characterized model organisms for investigation with environmental and clinical relevance²⁹⁻³¹.

We chose *E. coli* and *B. cereus* to demonstrate the possibility of antiviral resistance as a proof of principle. In the following lines we address how further study would be needed to investigate the effects of antivirals and the outcome of resistance on other bacteria or strains of the same species:

Lines 591-597: Given the diversity of gram-positive and gram-negative bacteria, using additional strains of *E. coli* and expanding this work beyond *E. coli* and *B. cereus* will help elucidate the broader implications for the effects of antibacterial antivirals on bacterial communities such as those found in environmental waters, wastewater treatment plants, and the human gut. Mechanisms of resistance to antiviral drugs are likely to vary across species and even strains of bacteria, and the patterns of cross-resistance that arise may therefore also be heterogeneous.

3. The phage aspect of the paper is very weak in terms of experiments. It is well known that growth has a huge impact of phage host interaction. Thus the observed effects could be due to change in growth. More over it is not precised at which multiplicity of infection the experiments

are performed (an absolute number of phage is given, but not MOI which is the relevant parameter), nor when the phages were added. Usually, phages are added during exponential growth at OD=0.3. Overall, I would suggest that the authors 1) Perform their experiments at different MOI by adding the phages in exponential phage growth. This would allow to also assess how strong the resistance is (might be for low MOI but not for high). 2) The authors confirm these results by doing similar experiments in solid media (plaque assays experiments).

Thank you for your comments. The phage results and discussion were removed from the manuscript upon revision, and we have added genetic analysis to streamline the results and discussion.

4. The cross resistance claim regarding antibiotics could also be more substantiated. While some effect was observable, it was sometimes more susceptible, sometimes more resistance, without any consistent observations in terms of mechanisms or modes of actions of antibiotics or antivirals. Could these be more “random” side effects? Also, maybe I missed it, but I did not see if all concentrations were tested and if yes, I did not find the results in the supp material, and how were the ones put forward chosen for main figure. If not, how were these chosen?

We thank the reviewer for all these questions and comments. We have substantiated the evidence by completing whole genome sequencing to investigate the unique mutations of each antiviral-resistant strain and corroborate the phenotypic results on resistance presented earlier. Several of the mutations identified in the antiviral-resistant isolates have been shown to have a role in antimicrobial resistance. Our future studies with additional strains and greater diversity of bacteria can hopefully further address whether we see patterns of cross-resistance, or whether cross-resistance seems to arise in a stochastic manner. The following text is included to describe antiviral-resistance mutations that occurred in genes known to be implicated in antimicrobial resistance:

Lines 384-396: Mutations in genes with a known role in antimicrobial resistance were observed in three cases. A mutation in the multidrug efflux pump, efflux resistance-nodulation-division (RND) transporter permease *acrB*, was observed in the dolutegravir-resistant *E. coli* isolate. In this unique case, the observed mutation led to a 1.8% increase in the length of the protein (Table 1). In didanosine-resistant *E. coli*, a deletion was observed in ABC transporter permease (single base pair deletion at position 3,338,728), suggesting potentially altered function to this protein family involved in the transport of a variety of substituents including not only carbohydrates, proteins, and lipids but also pharmaceutical drugs³⁸. The early stop codon introduced by the single base pair deletion led to a 35.2% reduction in protein length. Raltegravir-resistant and dolutegravir-resistant *E. coli* (both antiviral drugs from the integrase inhibitor class) exhibited the same mutation in cAMP-activated global transcriptional regulator (CRP), in which guanine was substituted for an adenine in *crp*. The substitution did not, however, lead to an alteration in the ultimate base pair length or amino acid sequence of the protein (Table S13).

From whole genome sequencing, we also detected mutation in thymidine kinase for zidovudine-resistant *E. coli* which has been reported in previous studies (e.g., Doléans-Jordheim 2011), which provides support that not all the mutations are random.

We have included the rationale for choosing the specific concentrations:

Line 255-258: The final concentrations of antibiotics used challenge wild type and mutant *E. coli* and *B. cereus* were selected after first challenging each individual strain and then choosing a concentration that captured the range of response to drug –i.e., any changes in growth-- across wild type and antiviral-resistant mutant strains for comparative analysis.

5. In the discussion, I feel the limits of the study were not properly discussed. The discussion is already very long, I would suggest to shorten it, especially as many of it is very speculative. While it's quite interesting to speculate and there can be space dedicated to it, first limits of the study should be discussed. I was typically wondering, when adding other types of bioactive compound (not anti-viral but of other types) how many would be expected to cause such effects that we observe? So are the authors in the study underlying a potential specific mechanism from anti-viral drugs, or just finding side effects of adding bioactive compound to cellular cultures?

Thank you for your suggestions. We have further discussed the limits of the study and shortened other aspects of the discussion. Your suggestion to investigate other types of bioactive compounds is an interesting one and certainly worthy of further comparative study. The limits are discussed as follows:

Lines 589-599: In the future, isolating and challenging a large number (>5) of antiviral-resistant isolates from each antiviral exposure will help answer whether antiviral-resistance mutations are consistent or stochastic. Given the diversity of gram-positive and gram-negative bacteria, using additional strains of *E. coli* and expanding this work beyond *E. coli* and *B. cereus* will help elucidate the broader implications for the effects of antibacterial antivirals on bacterial communities such as those found in environmental waters, wastewater treatment plants, and the human gut. Mechanisms of resistance to antiviral drugs are likely to vary across species and even strains of bacteria, and the patterns of cross-resistance that arise may therefore also be heterogeneous. The effects of mixtures of drugs --similar to those relevant in the human body or the highly heterogeneous context of wastewater—on bacteria also warrants further investigation.

6. Throughout the study, I would have enjoyed more thoughts/hypothesis on potential mechanisms that could explain the observed patterns of cross resistance (like the fact that most mechanisms led to broad resistance to other anti-viral compounds) and also given that mechanisms of anti viral actions and antibiotics are known what could be expected at the "crossroad".

Thank you for letting us know what we can add to improve the enjoyable nature of our study. We have added more thoughts and hypotheses on potential mechanisms as well as investigated them directly by sequencing isolates. The potential mechanisms leading to resistance are now mentioned in both the results and discussion section. The following lines mentioned previously hopefully help highlight some of the mutations that have been observed in antibiotic-resistant bacteria and were also observed in the antiviral-resistant isolates:

Lines 384-396: Mutations in genes with a known role in antimicrobial resistance were observed in three cases. A mutation in the multidrug efflux pump, efflux resistance-nodulation-division (RND) transporter permease *acrB*, was observed in the dolutegravir-resistant *E. coli* isolate. In this unique case, the observed mutation led to a 1.8% increase in the length of the protein (Table 1). In didanosine-resistant *E. coli*, a deletion was observed in ABC transporter permease (single base pair deletion at position 3,338,728), suggesting potentially altered function to this protein family involved in the transport of a variety of substituents including not only carbohydrates, proteins, and lipids but also pharmaceutical drugs³⁸. The early stop codon introduced by the single base pair deletion led to a 35.2% reduction in protein length. Raltegravir-resistant and dolutegravir-resistant *E. coli* (both antiviral drugs from the integrase inhibitor class) exhibited the same mutation in cAMP-activated global transcriptional regulator (CRP), in which guanine was substituted for an adenine in *crp*. The substitution did not, however, lead to an alteration in the ultimate base pair length or amino acid sequence of the protein (Table S13).

Minor points

1. While I am not familiar with anti-viral drugs, which might be the case of many readers of the study, it would be of help, in the introduction or in the text to give an order of magnitude at which each of these drugs are used, to estimate whether the tested concentrations are within interesting biological range.

Thank you for your suggestion. We added the following lines to discuss the clinical/biological implications of the concentrations tested:

Lines 132-141: The concentration range tested covers inhibitory and sub-inhibitory concentrations demonstrated in previous *in vitro* tests with bacteria¹⁰⁻¹². We included concentrations tested in existing literature to ensure reproducible results and to validate our methods. The concentration range tested also includes therapeutic or circulating concentrations of antiviral drugs, such as the plasma concentration of zidovudine (0.016-1.7 mg/L)³² and therapeutic window of efavirenz (1-4 mg/L)³³, as well as concentrations of antivirals detected in the aquatic environment that can reach $\mu\text{g/L}$ levels³⁴. However, in attempt to establish inhibitory values for various incubation periods and exposure concentrations, the concentration ranges tested generally exceeded environmentally detected levels or a typical plasma concentration in the human body³⁵.

2. I found the growth curves presented in S2 quite more informative than the summary of Figure 1. Maybe to keep the summary of figure 1 but adding to that figure, key growth curves. For example the one with zidobuvine was very convincing while some of the differences for others were much moe subtle / less conclusive

We thank the reviewer for this suggestion. We have added key growth curves to Figure 1 and modified the text of the figure caption:

Lines 169-180: Figure 1: (A) Antibacterial effects of antivirals after 4-24 hours of exposure at different antiviral concentrations ranging from 0.1 – 100 µg/mL. Nucleoside analogs acyclovir, didanosine, lamivudine, stavudine and zidovudine impact the growth of *E. coli* only. Both *E. coli* and *B. cereus* are significantly inhibited (with at least 10% reduction in growth compared to control) by non-nucleoside reverse transcriptase inhibitor efavirenz and the integrase inhibitors dolutegravir and raltegravir at concentrations from 10 -100 µg/mL beginning at four hours of exposure time for *E. coli* and eight hours for *B. cereus*. (B) Example of growth curves for *E. coli* treated with 0.1 – 100 µg/mL acyclovir over 24 hours. The most significant difference in growth, as observed in the growth curve and noted in the heat map (A) is only from 8-24 hours with 100 µg/mL acyclovir. (C) Example of growth curves for *E. coli* treated with 0.1 – 100 µg/mL zidovudine over 24 hours. Significant differences between untreated and zidovudine-treated *E. coli* are observable at all concentrations tested and across all time points from 4-24 hours.

3. Figure S2: experiments were done in triplicates, but what is represented there ? One out of the three experiments ? Means ? In any case, it would be good to get a sense of the variation (maybe each curve could have a shadow representing the variations between the replicates). Also n=3, means biological replicates ?

We thank the reviewer for asking these questions so that we can further clarify our data for the reader. First, what is represented in Figure S2 are raw absorbance values (y-axis) plotted over time (x-axis). These are technical replicates in which the variation in results are represented by the points and error bars. We have clarified what is represented in the figure legend, as follows:

Each graph represents data from a separate trial, n=3 technical replicates with the means plotted and error bars demonstrating variation among replicates.

An absence of observable error bars would indicate data that is very closely aligned. We have included this point in the caption of Figure 4, as follows:

Lines 357-358: ...absence of observable error bars at each data point are indicative of close alignment of triplicate measurements.

4. Figure S5: It is not explained what is “color key”. Not clear to me what red or green is. Also not explained how the hierarchical clustering was done.

Figure S5 has been updated with a label on the color key and an explanation of how the hierarchical clustering was done in the figure legend. The text reads as follows:

Mutant and wild-type *E. coli* and *B. cereus* strains were clustered according to their antiviral-resistance profiles. Hierarchical clustering was performed in R (v.3.5.3) using the heatmap.2 function without data scaling in the gplots package (v.3.0.1).

5. Figure 3: did not really understand how the statistics were performed

The statistical analyses are described in the Methods section, “4.5 Data Analysis” and included in Tables S9 and S10. We have ensured that we have described how the statistics are performed for Figure 3 in the following text:

Lines 702-708: The percent growth of antibiotic-treated bacteria compared to untreated bacteria was calculated based on averaging the raw absorbance values at the 16-hour time point across triplicate conditions for all strains and wild type tested. Minimum and maximum ranges for antibiotic effects were calculated by comparing the minimum and maximum absorbance values from triplicate conditions of antibiotic-treated and untreated wells. Percent growth relative to untreated control was calculated using Excel (Table S10).

6. There was no specific method section dedicated to phage, so could not assess it. How was it amplified ? titrated ? When were the phages added to growing bacteria ?...

The phage results and discussion were removed from the manuscript upon revision in order to make space for the whole genome analyses and streamline the results and discussion.

Reviewers' comments:

Reviewer #1 (Remarks to the Author):

The authors have altered the manuscript according to the suggestions given. I recommend this work for acceptance. However, I would like to suggest the authors to consider CFU measurement in the future as a way to gauge viability, as OD measurement is often an unreliable tool for that, although it might still be valid in certain circumstances. It would be preferable to use OD to gauge bacterial growth and CFU count for viability. Additionally, I would like to suggest the authors to explore more the phage part of the manuscript that has been eliminated in this revision, maybe in another study and taking into account all the comments given by both reviewers, as I believe they will be helpful to further develop this noteworthy topic.

Reviewer #3 (Remarks to the Author, also attached):

I got the opportunity to review the revised version of the manuscript entitled "Bacteria exposed to antiviral drugs develop antibiotic cross-resistance and unique resistance profiles". I understand that the previous version of the manuscript was reviewed by another reviewer, and now I have been assigned to review the revised paper.

Overall, this paper titled "Bacteria exposed to antiviral drugs develop antibiotic cross-resistance and unique resistance profiles" presents important findings that have significant implications for public health. The study aims to investigate whether antiviral drugs have the potential to induce antimicrobial resistance in bacteria and the results demonstrate that exposure to certain antiviral drugs can lead to the development of antibiotic cross-resistance in both gram-negative and gram-positive bacteria and the development of unique resistance profiles. After carefully reading the manuscript, I commend the authors for addressing the major issues raised by the previous reviewer and for providing additional data to support their findings. The revised manuscript is well-written and organized.

I have the following comments/suggestions:

Major Comments

1. The authors have conducted a genome analysis to identify the mechanisms of antimicrobial resistance (AMR). Here, authors have made certain claims which seem to be completely redundant. For example, mutations leading to a length increment or mutations in genes related to metabolic pathways. What is the usage for identifying a mutation in such random genes?
The evolution of genes is a common process which gives rise to different homologs. Particularly, orthologs differ significantly by the sequence similarity/identity, but not necessarily suggesting the alteration in gene functions. Given that the gene-function relationship depends on different parameters, gene sequence \diamond Protein sequence \diamond protein structure.
Given the concept of genome evolution for genome plasticity, how authors concluded that a mutation (particularly an unknown one) is leading to an alteration in gene functions but not a trade-off for survival? Stop codons or pseudogenization cause truncation, but how do indels lead to truncation in all identified genes (Table 1)? How have the authors concluded that the change in gene length is a truncation event?
2. Why was WGS performed on E. coli only?
3. Authors have carried out both genotypic and phenotypic analysis, thus WGS results must be presented in concordance and discordance with genotypic and phenotypic outputs.
4. Besides, the presentation of the WGS-related sections, including methodology, results, and discussion is not clear. For example,
 - Methodology to identify the antimicrobial resistance genes.
 - Methodology of mutation analysis.
 - If assembled followed by annotation, why gene coordinates are of assembly not of genes?
 - Table 1 needs more information viz, bacterial strain/ genome accession.

Overall, the WGS section seems a different section in the manuscript without proper analysis and

justification. This section has a lot of scopes and thus, it needs extensive revision in all parts (methodology, results, discussion, and conclusion).

5. Figure 1., how do authors interpret the results where the significance of antimicrobial effect has (a) increased with time, (b) decreased with time and (c) remain unchanged? What is the physical meaning and difference among $p < 0.005$, $p < 0.0005$, and $p < 0.00005$ If any results with $p < 0.05$ is significant or not, justify. Please discuss, the results.

6. Line 126: 150, please avoid putting references in the results section; discuss your results in the separately given discussion section.

Minor comments

Line 105: What do you mean by "more complete"?

Line 118: Correct the sentence "Results from...".

Line 138: can reach Levels? Complete it.

We have *italicized and underlined reviewer comments* on the manuscript. Our responses to reviewer comments are in order below the reviewer comment, in standard font style, with reference to the updated manuscript line number and updated manuscript text highlighted as blue font color.

REVIEWER #2

The authors have altered the manuscript according to the suggestions given. I recommend this work for acceptance. However, I would like to suggest the authors to consider CFU measurement in the future as a way to gauge viability, as OD measurement is often an unreliable tool for that, although it might still be valid in certain circumstances. It would be preferable to use OD to gauge bacterial growth and CFU count for viability. Additionally, I would like to suggest the authors to explore more the phage part of the manuscript that has been eliminated in this revision, maybe in another study and taking into account all the comments given by both reviewers, as I believe they will be helpful to further develop this noteworthy topic.

We thank the reviewer for carefully reviewing our revised manuscript and recommending it for acceptance. We appreciate the recommendation to consider CFU in the future in lieu of OD. We plan to pursue the bacteriophage aspects of the manuscript in future experiments for a separate manuscript, and we appreciate the reviewer's recommendation to continue to explore it.

REVIEWER #3

Major Comments:

The authors have conducted a genome analysis to identify the mechanisms of antimicrobial resistance (AMR). Here, authors have made certain claims which seem to be completely redundant. For example, mutations leading to a length increment or mutations in genes related to metabolic pathways. What is the usage for identifying a mutation in such random genes? The evolution of genes is a common process which gives rise to different homologs. Particularly, orthologs differ significantly by the sequence similarity/identity, but not necessarily suggesting the alteration in gene functions. Given that the gene-function relationship depends on different parameters, gene sequence -> Protein sequence -> protein structure.

I (A). What is the usage for identifying a mutation in such random genes?

We thank the reviewer for pointing out that we should clarify the utility of identifying changes in genes that have no known association with antibiotic resistance. As there were several genetic changes within each antiviral-resistant genome, we previously focused our attention on the ones that resulted in the largest changes. However, as the reviewer has indicated, we cannot say how

mutations in many genes not previously implicated in antibiotic resistance might influence the isolates. Thus, we have limited our presentation to genes that are the most likely to be causative of the observed phenotype, because the mutations are found in known or putative antibiotic or antiviral drug resistance. Thus, we have modified Table 1 and the results to highlight genes known or likely to be involved in antibiotic resistance as follows:

Table 1: Genetic changes in antiviral-resistant *E. coli* genomes with known or suggestive roles in antimicrobial resistance: blue highlighting indicates known role in antimicrobial resistance; green highlighting indicates recently suggested role in resistance.

Antiviral-Resistant E. coli Genome⁺	Mutation*	Gene	Function/Characteristics	Potential Role in Antimicrobial Resistance	Percent Protein Truncated Due to Mutation
Didanosine-resistant [CP115968]	(232) L: CTG → R: CGA	ABC transporter permease ydhY	 • Transmembrane transport of substituents including carbohydrates, lipids, proteins, drugs⁵¹ 	 • ABC transporter family involved in drug efflux⁴³ 	35.2
	(47) S: AGGT → R: AGG (48) E: GAG → R: AGG (50) E-K: GAA-AAA → R: CGA (53) G: GGA → R: CGC	Galactofuranose-binding protein ytfQ	 • Galactofuranose transport during carbon scavenging conditions • Possible additional functions⁵² 	 • Certain SNPs predictive of resistance to subinhibitory concentrations of antibiotics⁵⁰ 	83.4
Dolutegravir-resistant [CP117044]	(non-coding region) → (1) M: ATG	Resistance-nodulation-division (RND) transporter permease acrB	 • Multidrug efflux pump³⁹⁻⁴² 	 • Mechanism for drug resistance³⁹⁻⁴² 	-1.8 [#]
	(19) Y: TAC → C: TGC	cAMP-activated global transcriptional regulator crp	 • Global transcriptional regulator^{26,53} 	 • Repression of the genes encoding the MdtEF multidrug efflux pump⁵⁴ 	0
	(77) S: TCA → STOP TAA	Phosphotransferase system (PTS) N-acetylgalactosamine-specific transporter subunit IIB agaV	 • Carbohydrate transport^{55,56} 	 • Associated with pan-tolerance to antimicrobials²⁶ 	47.6
	(292) A-E: GCC-GAA → A-A: GCC-GCG	Sensory domain of two-component sensor histidine kinase dcuS	 • Directs response to variety of environmental stimuli⁴⁶ 	 • May be involved in antibiotic resistance⁴⁶ 	44.1
Lamivudine-resistant [CP115179]	(108) C: TGT → V: GTA	Hexose-6-phosphate:phosphate antiporter uhpT	 • Transporter for phosphate and glucose-, fructose- or mannose-6-phosphate⁵⁷ 	 • Role in antibiotic resistance⁴⁷ 	73.1
	(27) E: GAA → (non-coding region)	Bifunctional ligase/repressor birA	 • Bifunctional biotin ligase and biotin operon repressor⁵⁸ 	 • Proposed target to treat multi-drug resistant infection^{59,60} 	90.5

Raltegravir-resistant [CP117043]	(19) Y: TAC → C: TGC	cAMP-activated global transcriptional regulator crp	• Global transcriptional regulator ^{26,53}	• Repression of the genes encoding the MdtEF multidrug efflux pump ⁵⁴	0
Stavudine-resistant [CP115180]	(75) V-E: GTT-GAA → V-N: GTG-AAC	tRNA ₁ ^{Val} (adenine ³⁷ -N ⁶)- methyltransferase yfiC	• Methylates adenine in position 37 of tRNA ₁ ^{Val} 61	• May be implicated in antimicrobial resistance ^{48,49}	52.0
Zidovudine-resistant [CP117469]	(54) T-I: ACC-ATT → T-L: ACA-TTA	Bifunctional aspartate kinase/homoserine dehydrogenase I thrA	• Involved in biosynthesis of amino acids, nucleotides ⁶²	• Potential target for multi-drug-resistant infection ⁶³	90.9

⁺Accession number listed below each genome. *⁺“Mutation” column lists first in parentheses the codon number in the wild type reference gene that is mutated compared to the antiviral-resistant strain, followed by the mutation described as the single letter amino acid abbreviation and sequence of the wild type and the single letter amino acid abbreviation and altered sequence in the antiviral-resistant mutant gene. [#]The protein increased in length by 1.8% due to the observed mutation.

We have also modified the results section focusing the discussion on those mutations with known resistance associations as follows:

Lines 422-450: After observing that antiviral-resistant isolates of *E. coli* and *B. cereus* have phenotypically unique changes in resistance to antibiotics compared to wild type, we conducted whole genome sequencing of antiviral-resistant *E. coli* strains with the goal of identifying genetic changes that may explain resistance to either antiviral drugs or antibiotics. Comparative genomics of *B. cereus* isolates could not be conducted due to contamination during reference strain library preparation and small number of mutant strains. Overall, there were as few as two and as many as 23 unique genetic changes in the antiviral-resistant *E. coli* mutant genomes as compared with the reference strain that could not be attributed to either a change in the reference during sequencing (differences found in all mutant genomes) or to a sequencing error (e.g., insertion of A within A homopolymer sequence) (Table S13). Mutations observed in three of the mutant strains (raltegravir-resistant *E. coli*, [CP117043]; dolutegravir-resistant *E. coli*, [CP117044]; didanosine-resistant *E. coli*, [CP115968]; Table 1) had genetic changes in genes with established roles in antibiotic resistance. Raltegravir-resistant and dolutegravir-resistant *E. coli* isolates –both integrase inhibitor resistant strains– have the same base change (A→G) leading to a non-synonymous amino acid change (Y→C) in the cAMP-activated global transcriptional regulator *crp* (Table 1), associated with resistance to macrolides, potentially explaining part of their acquired antibiotic resistance phenotype of these two antivirals from the same antiviral class. A mutation in the multidrug efflux pump, efflux resistance-nodulation-division (RND) transporter permease *acrB*, was observed in the dolutegravir-resistant *E. coli* isolate. In this case, the observed mutation led to a 1.8% increase in the length of the protein (Table 1) and may explain the observed resistance to tetracycline³⁹⁻⁴². In didanosine-resistant *E. coli*, a deletion was observed in ABC transporter permease *yhdY* (single base pair deletion in codon 232), suggesting potentially altered function to this protein family involved in the transport of a variety of substituents including not only carbohydrates, proteins, and lipids but also pharmaceutical drugs⁴³. The early stop codon introduced by the single base pair deletion led to a 35.2% reduction in protein length (Table 1). Additionally, two mutants exposed to nucleoside analogs (lamivudine-resistant *E. coli*, [CP115179], cytidine analog; zidovudine-resistant *E. coli*, [CP117469], thymidine analog) had different genetic changes within thymidine kinase gene (*tdk*), suggesting a role in nucleoside analog antiviral drug resistance (Table S13).

Finally, we have modified our discussion as follows:

Lines 797-806: Given the interest in identifying mutations that could be causative of the resistance phenotype, mutations in genes such as those identified in this study may indicate how non-antibiotic drugs may still have antimicrobial effects and from where broad antimicrobial resistance may arise. In addition, the genes identified in this study may point to pathways that may be involved in resistance and therefore serve as targets for novel therapeutics beyond traditional antibiotics. The mutations identified in antiviral-resistant genomes may help explain antibiotic cross-resistance phenotypes due to antiviral exposure. The identified mutations may also provide further support that mutations in genes beyond traditional resistance genes may be implicated in a range of antimicrobial resistance.

1 (B). The evolution of genes is a common process which gives rise to different homologs. Particularly, orthologs differ significantly by the sequence similarity/identity, but not necessarily suggesting the alteration in gene functions. Given that the gene-function relationship depends on different parameters, gene sequence -> Protein sequence -> protein structure.

The reviewer brings up a good point, that we do not know how changes in a gene that cause changes in a protein sequence may influence the function. The function of the altered protein could still be homologous to the original. However, the necessary analysis of protein structure or additional experiments to verify the genetic changes as causative of phenotype are beyond the scope of this study. We focus our attention on establishing the relationship between antiviral drugs and antibiotic resistance, and we complete the genetic analysis to provide a starting point for future analyses. We have emphasized this point as:

Line 584-588: Although more work is needed to determine which of the genetic changes observed in the mutant genomes are important for the acquired antiviral or antimicrobial phenotypes, we have presented several likely candidates with a known role in antibiotic resistance, and which have led to the largest changes in protein sequences.

1 (C). Given the concept of genome evolution for genome plasticity, how authors concluded that a mutation (particularly an unknown one) is leading to an alteration in gene functions but not a trade-off for survival?

We appreciate the point the reviewer makes, that we don't know whether the mutations change function. We can only hypothesize that one or more mutations have led to the observed change in phenotype, in which we observe antibiotic cross-resistance. We have included the following additional details and references in the manuscript to address this point:

Lines 849-855: The ability of bacteria to adapt to environmental stressors including drugs may lead to many different alterations and adaptations for survival^{84,85}. Alterations in gene function can be both mechanisms to survive an environmental challenge and lead to or be associated with antimicrobial resistance^{53,77,86}. Antiviral drugs may have similar effects as stressors leading to resistance phenotypes, although further study is necessary for establishing the pathways and mechanisms by which antiviral drugs affect bacteria.

1 (D). Stop codons or pseudogenization cause truncation, but how do indels lead to truncation in all identified genes (Table 1)?

We have included more information in the following text to address this question:

Lines 1069-1072: For all mutations reported in Table 1, stop codons were observed after the mutation due to a frame shift in the gene introduced by the mutation. Stop codons were identified by observing the three sequential nucleic acids TAG, TAA, or TGA.

1 (E). How have the authors concluded that the change in gene length is a truncation event?

We have included the following additional information in the methods section to explain how truncation was determined:

Lines 1060-1069: Translations of amino acid sequences were observed in SnapGene Viewer v.6.2, and protein identity and sequence were confirmed again using blastx (<https://blast.ncbi.nlm.nih.gov/Blast.cgi>) entering the FASTA sequence and restricting the search to *E. coli* for organism (taxid:562). Truncation events were identified in antiviral-resistant genomes when, in comparison to wild type *E. coli B* and sequence matches in blastx, the antiviral-resistant gene was shortened due to an early stop codon following an identified mutation. Percent protein truncated due to mutation, reported in Table 1, was calculated based on comparison to wild type *E. coli B* gene length after the stop codon in the antiviral-resistant genome.

2. Why was WGS performed on *E. coli* only?

Whole genome sequencing (WGS) was performed in both *B. cereus* and *E. coli*; however, the *B. cereus* reference strain was contaminated, and so few isolates were obtained that the comparative analysis would be difficult. We have modified the text to explain this for the reader:

Lines 425-427: Comparative genomics of *B. cereus* isolates could not be conducted due to contamination during reference strain library preparation and small number of mutant strains.

3. Authors have carried out both genotypic and phenotypic analysis, thus WGS results must be presented in concordance and discordance with genotypic and phenotypic outputs.

We appreciate the reviewer's comment that the whole genome sequencing results should be better integrated with the phenotypic results. We have modified the beginning of the whole genome sequencing results section to improve this:

Lines 431-450: Mutations observed in three of the mutant strains (raltegravir-resistant *E. coli*, [CP117043]; dolutegravir-resistant *E. coli*, [CP117044]; didanosine-resistant *E. coli*, [CP115968]; Table 1) had genetic changes in genes with established roles in antibiotic resistance. Raltegravir-resistant and dolutegravir-resistant *E. coli* isolates –both integrase inhibitor resistant strains– have the same base change (A→G) leading to a non-synonymous amino acid change (Y→C) in the cAMP-activated global transcriptional regulator *crp* (Table 1), associated with resistance to macrolides, potentially explaining part of their acquired antibiotic resistance phenotype of these two antivirals from the same antiviral class. A mutation in the multidrug efflux pump, efflux resistance-nodulation-division (RND) transporter permease *acrB*, was observed in the dolutegravir-resistant *E. coli* isolate. In this case, the observed mutation led to a 1.8% increase in the length of the protein (Table 1) and may explain the observed resistance to tetracycline^{39–42}. In didanosine-resistant *E. coli*, a deletion was observed in ABC transporter permease *yhdY* (single base pair deletion in codon 232), suggesting potentially altered function to this protein family involved in the transport of a variety of substituents including not only carbohydrates, proteins, and lipids but also pharmaceutical drugs⁴³. The early stop codon introduced by the single base pair deletion led to a 35.2% reduction in protein length (Table 1). Additionally, two mutants exposed to nucleoside analogs (lamivudine-resistant *E. coli*, [CP115179], cytidine analog; zidovudine-resistant *E. coli*, [CP117469], thymidine analog) had different genetic changes within thymidine kinase gene (*tdk*), suggesting a role in nucleoside analog antiviral drug resistance (Table S13).

4. Besides, the presentation of the WGS-related sections, including methodology, results, and discussion is not clear. For example,

We thank the reviewer for recommending that we clarify the methodology, results and discussion. We have addressed each point below and included the modified text:

(A) • Methodology to identify the antimicrobial resistance genes.

Lines 1072-1078: The on-line Resistance Gene Identifier (RGI 6.0.1; <https://card.mcmaster.ca/analyze/rgi>) was used to identify genes within reference and mutant genomes matching the Comprehensive Antibiotic Resistance Database (CARD 3.2.6) restricting to perfect and strict matching criteria. Differences in gene sequences within antibiotic resistance genes was identified using a custom perl script. We also searched the literature for evidence of a role in antibiotic resistance for genes that contained the most significant changes to the protein coding genes.

(B) • Methodology of mutation analysis.

Lines 1054-1060: Mummer (4.0beta2) and DNASTAR MegAlign Pro (Version: 17.4.2) were used to identify genetic differences between wild type and antiviral-resistant genomes. Any base pair differences among the antiviral-resistant genomes compared to wild type were highlighted by the analysis. DNASTAR MegAlign Pro and SnapGene Viewer (version 6.1.1) were then used to cross-check each identified base pair mutation, comparing the sequence of wild type and antiviral-resistant genomes.

Lines 1060-1064: Translations of amino acid sequences were observed in SnapGene Viewer, and protein identity and sequence for the gene containing the mutation were confirmed again using blastx (<https://blast.ncbi.nlm.nih.gov/Blast.cgi>) entering the FASTA sequence and restricting the search to *E. coli* for organism (taxid:562).

(C) • If assembled followed by annotation, why gene coordinates are of assembly not of genes?

Lines 1049-1054: Custom Perl scripts and BLAST (version 2.13.0+) were used to adjust the starting position and orientation of each annotated antiviral-resistant genome, as the starting positions and orientations for each genome were not identical in the raw data. The gene coordinates were then numbered based on the unified starting position for all genomes. Codon numbers within each gene (identified using SnapGene Viewer) were used to describe the location of the identified mutations listed in Table 1.

(D) • Table 1 needs more information viz, bacterial strain/ genome accession.

Table 1 (starting on Line 591) has been modified accordingly to include the genome accession number, highlighted in blue. The captions above and below the table was also modified to describe the revised table:

Table 1: Genetic changes in antiviral-resistant *E. coli* genomes with known or suggestive roles in antimicrobial resistance: blue highlighting indicates known role in antimicrobial resistance; green highlighting indicates recently suggested role in resistance.

Antiviral-Resistant E. coli Genome⁺	Mutation*
Didanosine-resistant [CP115968]	(232) L: CTG → R: CGA
	(47) S: AGGT → R: AGG
	(48) E: GAG → R: AGG
	(50) E-K: GAA-AAA → R: CGA (53) G: GGA → R: CGC
Dolutegravir-resistant [CP117044]	(non-coding region) → (1) M: ATG
	(19) Y: TAC → C: TGC
	(77) S: TCA → STOP TAA
	(292) A-E: GCC-GAA → A-A: GCC-GCG
Lamivudine-resistant [CP115179]	(108) C: TGT → V: GTA
	(27) E: GAA → (non-coding region)
Raltegravir-resistant [CP117043]	(19) Y: TAC → C: TGC
Stavudine-resistant [CP115180]	(75) V-E: GTT-GAA → V-N: GTG-AAC
Zidovudine-resistant [CP117469]	(54) T-I: ACC-ATT → T-L: ACA-TTA

⁺Accession number listed below each genome. *⁺“Mutation” column lists first in parentheses the codon number in the wild type reference gene that is mutated compared to the antiviral-resistant strain, followed by the mutation described as the single letter amino acid abbreviation and

Overall, the WGS section seems a different section in the manuscript without proper analysis and justification. This section has a lot of scopes and thus, it needs extensive revision in all parts (methodology, results, discussion, and conclusion).

We thank the reviewer for the specific comments and recommendations on how to improve the methodology, results, discussion and conclusions of the manuscript. Some of the changes we have made in order to improve the manuscript are highlighted as follows:

Methods:

Lines 1040-1080: Assembly statistics confirmed a single contig for each genome sequenced and the total length of each genome as expected for *E. coli* (approximately 4.6 million base pairs) (Table S12). Comparative analysis of the sequenced genomes revealed unique single base pair insertions, deletions and substitutions, as well as multi-base pair insertions and deletions relative to the wild-type strains (Table 1, Table S13). Some mutations were found to be conserved across all sequenced antiviral-resistant mutants, which we concluded were collectively a unique mutation event in the wild-type clone during selection for sequencing (Table S13). All mutations in each antiviral-resistant *E. coli* genome were identified based on comparative analysis with *E. coli B* wild type isolate sequenced [accession number CP11578] in the same run as the antiviral-resistant strains. Custom Perl scripts and BLAST (version 2.13.0+) were used to adjust the starting position and orientation of each annotated antiviral-resistant genome, as the starting positions and orientations for each genome were not identical in the raw data. The gene coordinates were then numbered based on the unified starting position for all genomes. Mummer (4.0beta2) and DNASTAR MegAlign Pro (Version: 17.4.2) were used to identify genetic differences between wild type and antiviral-resistant genomes. Any base pair differences among the antiviral-resistant genomes compared to wild type were highlighted by the analysis. DNASTAR MegAlign Pro and SnapGene Viewer (version 6.1.1) were then used to cross-check each identified base pair mutation, comparing the sequence of wild type and antiviral-resistant genomes. Translations of amino acid sequences were observed in SnapGene Viewer, and protein identity and sequence for the gene containing the mutation were confirmed again using blastx (<https://blast.ncbi.nlm.nih.gov/Blast.cgi>) entering the FASTA sequence and restricting the search to *E. coli* for organism (taxid:562). Truncation events were identified in antiviral-resistant genomes when, in comparison to wild type *E. coli B* as well as nucleotide sequences from other *E. coli* (taxid:562) deposited to GenBank and retrieved using blastx, the antiviral-resistant gene was shortened after an early stop codon following an identified mutation. Percent protein truncated due to mutation, reported in Table 1, was calculated based on comparison to wild type *E. coli B* gene length after the stop codon in the antiviral-resistant genome. For all mutations reported in Table 1, stop codons were observed after the mutation due to a change in sequence of the gene introduced by the mutation. Stop codons were identified by observing the three sequential nucleic acids TAG, TAA, or TGA. The on-line Resistance Gene Identifier (RGI 6.0.1; <https://card.mcmaster.ca/analyze/rgi>) was used to identify genes within reference and mutant genomes matching the Comprehensive Antibiotic Resistance Database (CARD 3.2.6) restricting to perfect and strict matching criteria. Differences in gene sequences within antibiotic resistance genes was identified using a custom perl script. We also searched the literature for evidence of a

role in antibiotic resistance for genes that contained the most significant changes to the protein coding genes. All scripts used to compare the genomes of *E. coli* B wild type and antiviral-resistant *E. coli* strains are freely available (https://github.com/spacocha/mutant_strain_comparative_genomics).

Results:

Lines 422-450: After observing that antiviral-resistant isolates of *E. coli* and *B. cereus* have phenotypically unique changes in resistance to antibiotics compared to wild type, we conducted whole genome sequencing of antiviral-resistant *E. coli* strains with the goal of identifying genetic changes that may explain resistance to either antiviral drugs or antibiotics. Comparative genomics of *B. cereus* isolates could not be conducted due to contamination during reference strain library preparation and small number of mutant strains. Overall, there were as few as two and as many as 23 unique genetic changes in the antiviral-resistant *E. coli* mutant genomes as compared with the reference strain that could not be attributed to either a change in the reference during sequencing (differences found in all mutant genomes) or to a sequencing error (e.g., insertion of A within A homopolymer sequence) (Table S13). Mutations observed in three of the mutant strains (raltegravir-resistant *E. coli*, [CP117043]; dolutegravir-resistant *E. coli*, [CP117044]; didanosine-resistant *E. coli*, [CP115968]; Table 1) had genetic changes in genes with established roles in antibiotic resistance. Raltegravir-resistant and dolutegravir-resistant *E. coli* isolates –both integrase inhibitor resistant strains– have the same base change (A→G) leading to a non-synonymous amino acid change (Y→C) in the cAMP-activated global transcriptional regulator *crp* (Table 1), associated with resistance to macrolides, potentially explaining part of their acquired antibiotic resistance phenotype of these two antivirals from the same antiviral class. A mutation in the multidrug efflux pump, efflux resistance-nodulation-division (RND) transporter permease *acrB*, was observed in the dolutegravir-resistant *E. coli* isolate. In this case, the observed mutation led to a 1.8% increase in the length of the protein (Table 1) and may explain the observed resistance to tetracycline^{39–42}. In didanosine-resistant *E. coli*, a deletion was observed in ABC transporter permease *yhdY* (single base pair deletion in codon 232), suggesting potentially altered function to this protein family involved in the transport of a variety of substituents including not only carbohydrates, proteins, and lipids but also pharmaceutical drugs⁴³. The early stop codon introduced by the single base pair deletion led to a 35.2% reduction in protein length (Table 1). Additionally, two mutants exposed to nucleoside analogs (lamivudine-resistant *E. coli*, [CP115179], cytidine analog; zidovudine-resistant *E. coli*, [CP117469], thymidine analog) had different genetic changes within thymidine kinase gene (*tdk*), suggesting a role in nucleoside analog antiviral drug resistance (Table S13).

Discussion and Conclusions:

Lines 765-806: While the whole genome sequencing results obtained cannot directly explain the phenotypic resistance profiles without further in-depth biological investigation, the results hint at pathways and targets that may be involved in resistance caused by exposure to antiviral drugs. Two out of the six *E. coli* isolates with a mutation in the same gene (thymidine kinase gene *tdk*) suggest that altered nucleoside metabolism could be an important mechanism to protect against the antimicrobial effects of nucleoside analogs. Thymidine kinase phosphorylates

thymidine and deoxyuridine, both of which are involved in numerous metabolic pathways in *E. coli*^{44,45}.

Two out of the six *E. coli* isolates also had the same genetic change in a global transcriptional regulator, *crp* also suggesting that gene regulation may play an important role. Also of note was the observed mutation in tRNA_{I^{Val}} (adenine³⁷-N⁶)-methyltransferase in stavudine-resistant *E. coli*. Recent studies have pointed to the role of tRNA modification genes in resistance to numerous classes of antibiotics^{48,49}. Finally, mutations in several transporters were identified -- including the ABC transporter permease *yhdY*, the PTS N-acetylgalactosamine-specific transporter *agaV*, the hexose-6-phosphate:phosphate antiporter *uhpT* and ABC transporter galactofuranose-binding protein *ytfQ* -- adding to the important role of membrane processes in antibiotic resistance mechanisms.

Although several of the observed genetic mutations in antiviral-resistant isolates occurred in proteins or pathways with known or suspected roles in antimicrobial resistance, the majority of genetic changes occurred in genes that are not known to be involved in antibiotic resistance. Interestingly, many of the genes in which the mutations occurred are known to be involved in aspects of metabolism and nutrient transport such as galactofuranose-binding protein, sodium/glutamate transporter, and acetate coenzyme A-transferase. Consistent with the existing literature that suggest an array of metabolic pathways may be implicated in antimicrobial resistance or may provide targets for novel therapies for multidrug-resistant bacterial infections^{26,46,59,60,63}, these mutations may still point to key genes involved in resistance to antiviral drugs and associated antibiotic cross-resistance. Many genes beyond those traditionally thought to be responsible for antibiotic resistance have been highlighted as relevant for clinical isolates with antibiotic resistance, such as mutations in N-acetylglycosamine deacetylase⁷⁴, glycerol kinase²⁷, glycosyltransferase⁷⁵, serine-threonine kinase⁷⁶, and histidine kinase⁷⁷. Given the interest in identifying mutations that could be causative of the resistance phenotype, mutations in genes such as those identified in this study may indicate how non-antibiotic drugs may still have antimicrobial effects and from where broad antimicrobial resistance may arise. In addition, the genes identified in this study may point to pathways that may be involved in resistance and therefore serve as targets for novel therapeutics beyond traditional antibiotics. The mutations identified in antiviral-resistant genomes may help explain antibiotic cross-resistance phenotypes due to antiviral exposure. The identified mutations may also provide further support that mutations in genes beyond traditional resistance genes may be implicated in a range of antimicrobial resistance.

5 (A). Figure 1., how do authors interpret the results where the significance of antimicrobial effect has (a) increased with time, (b) decreased with time and (c) remain unchanged?

We thank the reviewer for suggesting that we improve our description of the significance of these effects. We have modified the text as follows to address the comment:

Lines 150-151: In some cases, the significance of antimicrobial effect was maintained over time, while in other cases it changed over time, either increasing or decreasing (Fig. 1).

Lines 755-759: Further investigation may help illuminate where and how the antiviral drugs exert their antimicrobial influence in cell replication or metabolism and how growth conditions also impact the effects of antimicrobials on bacteria³⁷. The significance level of antimicrobial effect may help indicate that some drugs act early in cell killing while others only take effect after more extended incubation.

5 (B). What is the physical meaning and difference among $p < 0.005$, $p < 0.0005$, and $p < 0.00005$ If any results with $p < 0.05$ is significant or not, justify. Please discuss, the results.

Thank you for suggesting we clarify the differences between the levels of significance described in Figure 1. We have modified the text as follows and included the results without significance in the results section:

Lines 151-210: We were conservative with what we considered a significant inhibition of growth, and we required both significant ($p < 0.005$) and substantial ($>10\%$) reduction in growth of the antiviral-treated bacteria compared to untreated controls (Table S3, Fig. S2). Differences were determined significant if the p-value of the t-test met the threshold ($p < 0.005$). All significant differences were further differentiated according to their degree of significance (from most to least significant: $p < 0.00005$, $p < 0.0005$, $p < 0.005$) for comparative purposes. Results that were not significant were interpreted as no significant difference in growth between the drug-treated condition and the untreated condition.

6. Line 126: 150, please avoid putting references in the results section; discuss your results in the separately given discussion section.

We appreciate the recommendation to avoid references in the results section. In response to a previous reviewer's comment on the original manuscript, we placed references in lines 126-150 to improve the explanation and justification of the methods that led to the results. We have modified the text by moving most of the material containing references to the discussion and methods sections as follows:

Lines 720-731: (**Discussion**): Gram-negative *E. coli* and gram-positive *B. cereus* were chosen as well-characterized model organisms for investigation with environmental and clinical relevance²⁹⁻³¹. Antivirals were tested at concentrations ranging from 0.1 – 100 $\mu\text{g/mL}$ over 0 – 24 hours (Fig. 1). The concentration range tested covers inhibitory and sub-inhibitory concentrations demonstrated in previous *in vitro* tests with bacteria¹⁰⁻¹². We included concentrations tested in existing literature to ensure reproducible results and to validate our methods. The concentration range tested also includes therapeutic or circulating concentrations of antiviral drugs, such as the plasma concentration of zidovudine (0.016-1.7 mg/L)³² and therapeutic window of efavirenz (1-4 mg/L)³³, as well as concentrations of antivirals detected in the aquatic environment that can reach concentrations in the $\mu\text{g/L}$ range³⁴. However, in attempt to establish inhibitory values for various incubation periods and exposure concentrations, the concentration ranges tested generally exceeded environmentally detected levels or a typical plasma concentration in the human body³⁵.

Lines 755-759: (**Discussion**): Further investigation may help illuminate where and how the antiviral drugs exert their antimicrobial influence in cell replication or metabolism and how growth conditions also impact the effects of antimicrobials on bacteria³⁷. The significance level of antimicrobial effect may help indicate that some drugs act early in cell killing while others only take effect after more extended incubation.

Lines 932-940: (**Methods**): Bacterial growth –measured as optical density (OD)-- was monitored at different time points in the growth curves of *E. coli* and *B. cereus* to better understand the influence of antiviral drugs through the phases of bacterial growth. OD is a common approach to monitoring bacterial growth in culture³⁶. We assessed the correlation between OD and colony forming units (CFUs) on untreated bacteria as well as across conditions challenging bacteria with both antiviral drugs with antibacterial properties and antiviral drugs with no putative antibacterial properties, and we found OD significantly correlated to CFU (Table S2). Since using OD allowed for measurements in high-throughput manner, we used OD (absorbance 600 nm) as an indicator of bacterial growth for monitoring the impact of antiviral drugs on bacteria in culture.

Minor Comments:

1. Line 105: What do you mean by “more complete”?

We have revised the sentence in line 105 to clarify the meaning of “more complete”:

Lines 104-107: Therefore, examining both the phenotypic and genotypic impacts of non-antibiotic drug exposure such as antivirals on bacteria can lead to **better** understanding of cross-resistance and the mechanisms leading to resistance.

2. Line 118: Correct the sentence “Results from...”.

We have corrected the grammar in the sentence of line 118 to read as follows:

Lines 119-122: The results of this study suggest that antiviral drugs with antibacterial properties can impact the antimicrobial resistance phenotype of bacteria; further investigation may help elucidate the specific mechanisms by which these drugs and potentially others may act on bacteria and contribute to antibiotic cross-resistance.

3. Line 138: can reach Levels? Complete it.

We have revised and completed the sentence as follows to clarify:

Lines 725-729: The concentration range tested also includes therapeutic or circulating concentrations of antiviral drugs, such as the plasma concentration of zidovudine (0.016-1.7

mg/L)³² and therapeutic window of efavirenz (1-4 mg/L)³³, as well as concentrations of antivirals detected in the aquatic environment that can reach concentrations in the µg/L range³⁴.

REVIEWERS' COMMENTS:

Reviewer #3 (Remarks to the Author):

First of all, I want to take this opportunity to apologize for the delayed review. I appreciate your patience and understanding throughout this process.

I have reviewed the final revised version of the manuscript titled "Bacteria Exposed to Antiviral Drugs Develop Antibiotic Cross-Resistance and Unique Resistance Profiles," and based on the substantial improvements made, I highly recommend accepting the manuscript for publication.

The revisions the authors have undertaken demonstrate their dedication to addressing the comments and suggestions, resulting in a significantly enhanced manuscript. The clarity and coherence of the information have been notably improved, with a more cohesive and comprehensive flow throughout the paper.

The clarified methodology, results, and interpretation now provide a robust and detailed description of the experimental design, data analysis, and key findings. This ensures that readers can fully comprehend the significance of your research and its implications within the field.